# MITIGATING HALLUCINATION IN MULTIMODAL REASONING VIA FUNCTIONAL ATTENTION CONTROL

## ABSTRACT

Multimodal large reasoning models (MLRMs) are rapidly advancing vision-language reasoning and are emerging as a foundation for cross-modal intelligence. Hallucination remains a persistent failure mode, manifesting itself as erroneous reasoning chains and misinterpretation of visual content. In this study, we observe that attention heads exhibit a staged division: **shallow** heads predominantly serve perception, while **deeper** heads shift toward symbolic reasoning, revealing two major causes of hallucination, namely perceptual bias and reasoning drift. To address these issues, we propose a lightweight and interpretable two-step plugin, Functional Head Identification and Class-conditioned Rescaling, which locates perception- and reasoning-oriented heads and regulates their contributions without retraining. Evaluations on three real-world MLRMs (`Kimi-VL`, `Ocean-R1`, `R1-Onevision`), six benchmarks across three domains, and four baselines show that our plugin achieves an average improvement of **5%** and up to **15%**, with **only $<1\%$ additional computation** and **9%** of baseline latency. Our approach is completely model-agnostic and significantly enhances both the reliability and interpretability of the off-the-shelf MLRMs, thereby enabling their safe deployment in high-stakes applications. Our code is available at https://anonymous.4open.science/r/Functional-Attention-Control.

## 1 INTRODUCTION

Large language models (LLMs) (Team, 2024; Guo et al., 2025; Team et al., 2025) have rapidly transformed natural language processing, and more recently multimodal large reasoning models (MLRMs) (Xu et al., 2024; Ming et al., 2025; Bai et al., 2025) are extending this revolution to vision–language tasks. By integrating powerful language reasoning with visual understanding, MLRMs are becoming a cornerstone for cross-modal intelligence (Huang et al., 2023), with broad potential for advancing both research (Wang et al., 2024a) and real-world applications (Yin et al., 2024). Despite these advances, a fundamental barrier — hallucinations — still remains (Liu et al., 2024; Wu et al., 2025). Existing models frequently generate conclusions that either conflict with the visual evidence or contradict their own reasoning chains (Bai et al., 2024). Such failures not only diminish the reliability of individual predictions but also erode trust in the reasoning process itself (Ding et al., 2025), posing serious obstacles for deploying MLRMs in high-stakes (Sokol & Vogt, 2023) domains where accountability and interpretability are crucial.

For LLMs, hallucination phenomena are often attributed to limited knowledge coverage (Huang et al., 2025) or decoding bias (Rawte et al., 2023; Ji et al., 2023), and mitigation methods include retrieval-based correction (Lewis et al., 2020), contrastive decoding (Chuang et al., 2023), and self-consistency (Madaan et al., 2023). In multimodal settings, the causes of hallucinations are more complex. The prevailing view is that they mainly stem from *limited use of visual evidence*: MLRMs may overlook details during encoding or lose critical information during cross-modal alignment (Guan et al., 2024; Li et al., 2023b). As a result, most existing designs attempt to "compensate for vision" through stronger supervision (Sun et al., 2024; Liu et al., 2023; Xie et al., 2024), finer-grained alignment (Peng et al., 2023; Chen et al., 2023a), or the use of external visual priors (Zhao et al., 2025b; Kim et al., 2024). While these approaches reduce hallucinations to some extent, they share a common assumption: *the root cause lies mainly in under-utilized visual information*.

Building on this assumption, recent work (Huang et al., 2024; Kang et al., 2025a) reveals that hallucinations also involve **deeper allocation dynamics** inside the model. Interpretability studies (Bi

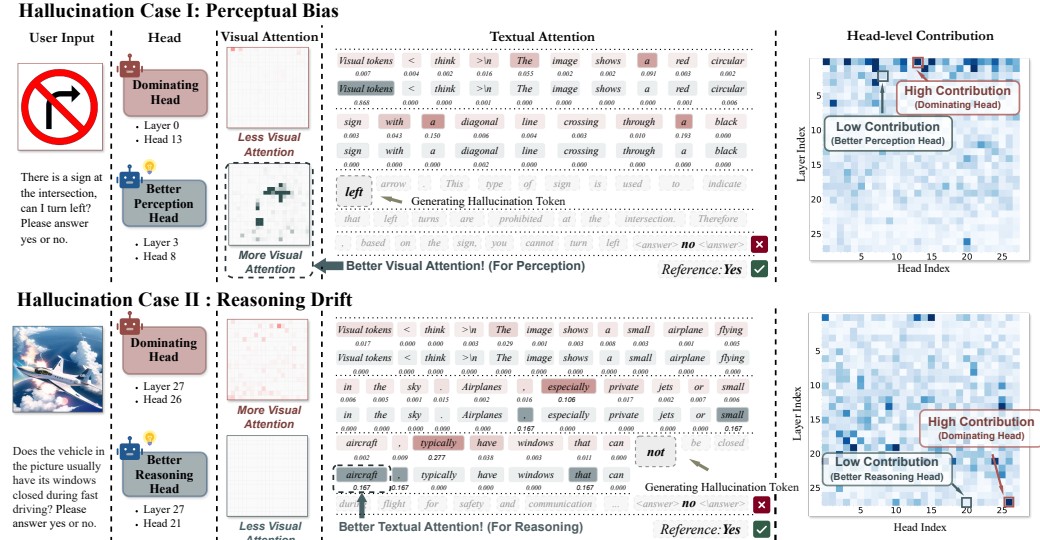

Figure 1: Two hallucination examples corresponding to perception layers (Cause I) and reasoning layers (Cause II). The figure highlights attention patterns over text and image tokens, and the contribution of tokens at the first position where hallucination emerges.

et al., 2025; Li et al., 2023a) show that MLRMs exhibit a staged division of attention: shallow layers rely on visual signals to extract evidence, while deeper layers increasingly shift toward symbolic reasoning over text. This staged allocation suggests that failures may arise not only from limited perception but also from misaligned usage, reflecting an imbalance in how vision and language are allocated and utilized across layers. Naturally, this raises a key research question: *under off-the-shelf architectures, how can we effectively identify and regulate the use of visual and textual information at different stages to mitigate hallucinations in MLRM's reasoning?*

To tackle this issue, we first decompose hallucinations in MLRMs into two primary failure causes. The first is ***Perceptual Bias*** (Pan et al., 2024; Zhao et al., 2025a), which occurs in *shallow layers* when attention over visual tokens becomes diffuse and critical evidence is diluted. The second is ***Reasoning Drift*** (Becker et al., 2025), which arises in *deeper layers* when attention fails to preserve intermediate steps, causing conclusions to deviate from established premises. To address these two causes, we proceed from the premise that models may already contain attention heads with the potential to support perception and reasoning (Jiang et al., 2025), but in their current form, these heads do not play a dominant role (Gong et al., 2024; Zhang et al., 2024a). Consequently, mitigating hallucinations requires identifying such heads and amplifying their contributions, so that perception and reasoning can be more effectively regulated across stages.

In this paper, we propose a lightweight and interpretable two-step plugin: ***Functional Head Identification*** and ***Class-Conditioned Rescaling***. ❖ The goal of ***Functional Head Identification*** is to isolate heads that naturally specialize in perception or reasoning, so that they can be explicitly leveraged rather than treated uniformly. To this end, we materialize attention weights, compute modality-specific attention ratios, and apply depth-aware boundaries to categorize heads into perception-oriented or reasoning-oriented groups. ❖ The goal of ***Class-Conditioned Rescaling*** is to amplify the contributions of these functional heads, thereby counteracting perceptual bias and reasoning drift without altering the underlying attention mechanism. This is achieved by assigning targeted multiplicative gains to the identified heads and rescaling their outputs in a structured yet minimally invasive manner.

**Experimental Takeaways.** We conduct systematic experiments on three real-world MLRMs (`Kimi-VL`, `Ocean-R1`, `R1-Onevision`), applying *Identification* and *Rescaling* to analyze their impact on perception and reasoning performance. In the ❖ **Identification Step**, we evaluate over 150 boundary configurations across perception and reasoning stages, finding a $27.4\% \uparrow$ between the best and worst settings, an observation that validates the necessity of stage-wise reasoning and highlights clear functional boundaries. During the ❖ **Rescaling Step**, we evaluate over 24 scaling strategies and observe that a moderate factor of 1.14 consistently achieved the best trade-off, under-

scoring that virtually all attention heads may carry latent contributions. ❖ **Overall Performance** indicates that across 6 benchmarks spanning three domains and against four baseline methods, our approach delivered an average improvement of 8% over the original MLRM, with gains reaching up to 20% in the most challenging tasks. Moreover, our scheme is *plug-and-play* without retraining, reaching SOTA performance with only $\sim 4\%$ extra computation, while introducing merely 7% of the latency compared to the best performance baseline.

## 2 MOTIVATION

### 2.1 MODALITY-INDEXED ATTENTION IN TRANSFORMERS

We begin by formalizing how Transformer attention distributes across modalities. Consider a sequence of $N$ tokens indexed by $\{1, \ldots, N\}$, partitioned into disjoint sets $\mathcal{T}_v$ (vision) and $\mathcal{T}_t$ (text), with $\mathcal{T}_v \cup \mathcal{T}_t = \{1, \ldots, N\}$ and $\mathcal{T}_v \cap \mathcal{T}_t = \varnothing$. Unless otherwise noted, we use the entire sequence as the query set, i.e., $\mathcal{T}_q = \{1, \ldots, N\}$.

For layer $\ell$ and head $h$, the standard multi-head attention mechanism computes

$$A^{(h,\ell)} = \text{Softmax}_{\text{row}}\left(\frac{(X^{(\ell)}W_Q^{(h,\ell)})(X^{(\ell)}W_K^{(h,\ell)})^\top}{\sqrt{d_k}}\right) \in \mathbb{R}^{N \times N}, \tag{1}$$

where $X^{(\ell)}$ denotes the input hidden states at layer $\ell$ ($N$ tokens, each of dimension $d$), $W_Q^{(h,\ell)}, W_K^{(h,\ell)}$ are the query and key projection matrices, and $d_k$ is the key dimension used for scaling. The value projection is $V^{(h,\ell)} = X^{(\ell)}W_V^{(h,\ell)}$, and the per-head output is $O^{(h,\ell)} = A^{(h,\ell)}V^{(h,\ell)}$, which are concatenated across heads and projected to yield the layer output $Y^{(\ell)}$.

### 2.2 PERCEPTION AND REASONING OF LMMS

Most large multimodal models (LMMs; e.g., `Qwen, LLaVA`) and their long-reasoning extensions (MLRMs; e.g., `Ocean-R1, Kimi`) are architecturally composed of a vision encoder coupled with a language model. This design naturally suggests a division of labor: the encoder **perceives** by compressing raw images into visual tokens, while the LM **reasons** by integrating those tokens with linguistic knowledge to produce answers. If this intuition is mirrored in internal computation, attention dynamics should follow the same workflow: *early layers emphasize visual tokens, whereas deeper layers progressively shift focus toward textual tokens*.

Many empirical studies support this view. An early analysis (Kang et al., 2025b; Zhang et al., 2024c) shows that pruning heads with high visual allocation leads to larger performance drops in vision-conditioned tasks than pruning random heads, with such "visual heads" especially common in shallow and middle layers. Complementarily, a subsequent study (Bi et al., 2025) reports a consistent layerwise trajectory: early layers show a high visual ratio with low concentration (broad scanning), middle layers keep a high ratio while concentration rises (focusing and cross-modal alignment), and late layers reduce the visual ratio as linguistic cues dominate. Together, the results indicate a stratified **perceive–then–reason** pipeline.

Motivated by these findings, we introduce a layerwise abstraction with two boundaries: $\ell_{\text{perc}}$ denotes the last layer where perception dominates, $\ell_{\text{reas}}$ denotes the first layer where reasoning dominates.

$$\mathcal{L}_{\text{perc}} = \{1, \ldots, \ell_{\text{perc}}\}, \quad \mathcal{L}_{\text{reas}} = \{\ell_{\text{reas}}, \ldots, L\}, \quad 1 \leq \ell_{\text{perc}}, \ell_{\text{reas}} \leq L, \quad \ell_{\text{perc}} \lesseqgtr \ell_{\text{reas}}. \tag{2}$$

In $\mathcal{L}_{\text{perc}}$ (**perception layers**), attention is biased toward visual tokens and gradually sharpens its focus, enabling the extraction and structuring of visual evidence. In $\mathcal{L}_{\text{reas}}$ (**reasoning layers**), attention shifts toward textual tokens, allowing linguistic context to dominate inference and generation. Notably, $\ell_{\text{perc}}$ and $\ell_{\text{reas}}$ need not coincide: the two sets may overlap (if some layers support both perception and reasoning) or leave a gap (if neither dominates in the intermediate depth).

This flexible abstraction captures the empirical transition from perception to reasoning. It also provides the basis for our hallucination analysis in Section 2.3.

## 2.3 HALLUCINATION IN PERCEPTION AND REASONING

Given the **perception–reasoning** pipeline established in Section 2.2, it is natural to analyze hallucination along these two stages. Figure 1 illustrates how failures in **perception** and **reasoning layers** give rise to two distinct but complementary forms of hallucination, which can be summarized as:

> ❖ *Cause I (Perceptual Bias).* Perception layers fail to provide sufficient visual evidence because it fails to allocate sufficient attention to the most informative regions.

In $\mathcal{L}_{\text{perc}}$, attention is expected to highlight salient cues so that later stages can build on accurate premises. However, we observe that dominant attention frequently under-represent critical regions, which may leaving the perceptual foundation incomplete and weakening subsequent reasoning.

> ❖ *Cause II (Reasoning Drift).* Reasoning layers fail to retain prior evidence, as attention often diffuses toward image tokens instead of focusing on critical steps in the reasoning chain.

In $\mathcal{L}_{\text{reas}}$, effective reasoning requires sustained attention to intermediate inference tokens that support the conclusion. Yet we find that dominating heads are sometimes distracted by residual visual signals, causing them to overlook core textual cues and thereby weakening the consolidation of prior evidence.

Taken together, *perceptual bias* and *reasoning drift* interact synergistically. Perception layers marginalize accurate visual cues before they reach the deeper network, while reasoning layers amplify irrelevant or weakly grounded signals. As illustrated in Figure 1, the compounding effect of these two failures greatly increases the likelihood of hallucination, leading to outputs that contradict the model's own intermediate evidence.

## 3 METHOD

Illustrated in Figure 2, we directly target the two root causes of hallucination identified in Section 2.3: *perceptual bias* in shallow and *reasoning drift* in deeper layers. The key idea is to reweight attention heads in a *depth-aware* manner, amplifying early heads that correctly capture visual evidence and late heads that preserve reasoning chains, while leaving the standard attention computation intact.

Concretely, the method proceeds in two stages. *(I) Functional Head Identification*: we explicitly materialize attention weights, compute the visual-attention ratio $S_v^{(\ell)}(h)$, and use the perception and reasoning boundaries $(\ell_{\text{perc}}, \ell_{\text{reas}})$ to identify perception- and reasoning-oriented heads. *(II) Class-conditioned Rescaling*: we assign multiplicative gains ($g_{\text{perc}}$ or $g_{\text{reas}}$), lift them to a head-aligned scaling vector, and apply this vector to rescale the concatenated head outputs.

### 3.1 FUNCTIONAL HEAD IDENTIFICATION

The first step of our method is to identify attention heads that function primarily as **perception heads** or **reasoning heads**, following the perceive–then–reason pipeline outlined in Section 2.2. This categorization enables targeted reweighting rather than treating all heads uniformly. For each layer $\ell$ and head $h$, we denote the attention matrix by $A^{(h,\ell)} \in \mathbb{R}^{N \times N}$. Its $(i, j)$-th entry is written as

$$a_{ij}^{(h,\ell)} = \left[ A^{(h,\ell)} \right]_{ij}, \qquad \sum_{j=1}^{N} a_{ij}^{(h,\ell)} = 1, \ \forall i. \tag{3}$$

Here, $a_{ij}^{(h,\ell)}$ is the normalized attention weight from query token $i$ to key token $j$. In other words, each row of $A^{(h,\ell)}$ forms a probability distribution over all keys, and $a_{ij}^{(h,\ell)}$ quantifies how much information query $i$ draws from token $j$.

To measure how head $h$ distributes attention across modalities, we compute the *modality attention ratio*. Using $\mathcal{T}_v$ and $\mathcal{T}_t$ to denote the sets of visual and textual tokens, and $\mathcal{T}_q$ as the set of query positions, we define the *modality attention ratio*:

$$S_m^{(\ell)}(h) = \frac{1}{|\mathcal{T}_q|} \sum_{i \in \mathcal{T}_q} \sum_{j \in \mathcal{T}_m} a_{ij}^{(h,\ell)}, \qquad m \in \{v, t\}. \tag{4}$$

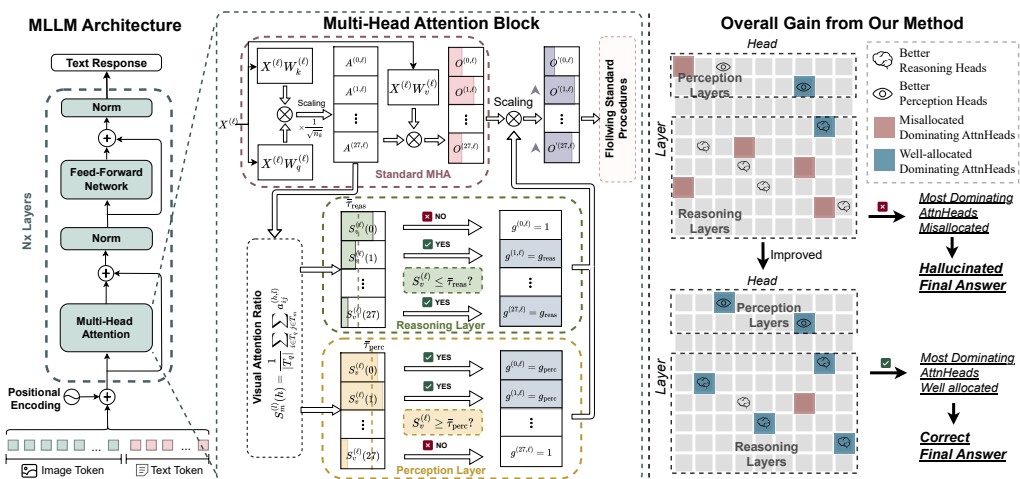

Figure 2: Overall architecture. The attention module is edited by computing the Visual Attention Ratio, which determines the rescaling of specific heads. This process promotes more effective heads to become dominating heads, guiding the output toward correct perception and reasoning. The softmax function is not explicitly shown in the figure.

where $m = v$ corresponds to visual and $m = t$ to textual tokens. Intuitively, $S_v^{(\ell)}(h)$ measures the fraction of attention mass allocated to visual tokens, averaged over all queries, while $S_t^{(\ell)}(h)$ does the same for textual tokens. Since $\mathcal{T}_v$ and $\mathcal{T}_t$ partition the sequence, we have $S_v^{(\ell)}(h) + S_t^{(\ell)}(h) = 1$. A larger $S_v^{(\ell)}(h)$ indicates *stronger visual focus*, while a smaller value indicates *stronger textual focus*.

We then combine these attention ratios with depth information to stratify heads. Specifically, we adopt two ratio thresholds, $\tau_{\text{perc}}$ and $\tau_{\text{reas}}$ with $\tau_{\text{reas}} < \tau_{\text{perc}}$, which distinguish heads that strongly attend to visual tokens (above $\tau_{\text{perc}}$) from those that focus on text (below $\tau_{\text{reas}}$). In addition, we use the perception and reasoning layer boundaries $\ell_{\text{perc}}, \ell_{\text{reas}}$ introduced in Section 2.2 to restrict these two head types to shallow and deep layers, respectively. Formally, for each layer $\ell$ we define

$$\mathcal{H}_{\text{perc}}^{(\ell)} = \big\{ h : \ell \leq \ell_{\text{perc}} \wedge S_v^{(\ell)}(h) \geq \tau_{\text{perc}} \big\}, \quad \mathcal{H}_{\text{reas}}^{(\ell)} = \big\{ h : \ell \geq \ell_{\text{reas}} \wedge S_v^{(\ell)}(h) \leq \tau_{\text{reas}} \big\}. \quad (5)$$

Heads that fall between thresholds or outside two-layer boundaries remain unlabeled.

Concretely, $\mathcal{H}_{\text{perc}}^{(\ell)}$ denotes shallow heads focusing on visual tokens, while $\mathcal{H}_{\text{reas}}^{(\ell)}$ represents deeper heads attending to textual tokens. At the layer level, the former strengthens visual grounding in perception layers ($\ell \leq \ell_{\text{perc}}$), and the latter reinforces inference consistency in reasoning layers ($\ell \geq \ell_{\text{reas}}$). Thus, enhancing perception heads mitigates **perceptual bias**, and enhancing reasoning heads counters **reasoning drift**, motivating our targeted rescaling strategy.

## 3.2 CLASS-CONDITIONED RESCALING

Building on Section 3.1, we target **perception-oriented heads in shallow layers** and **reasoning-oriented heads in deep layers**, denoted by $\mathcal{H}_{\text{perc}}^{(\ell)}$ and $\mathcal{H}_{\text{reas}}^{(\ell)}$.

Our guiding principle is to intervene on as few heads as possible while maximizing impact on the final outcome. Prior work (Nam et al., 2025; Michel et al., 2019), as well as our case analysis in Figure 2, shows that only a small number of *dominating heads* exert decisive influence on reasoning, while the majority have limited effect. Consequently, the most effective strategy is to increase the likelihood that members of $\mathcal{H}_{\text{perc}}^{(\ell)}$ and $\mathcal{H}_{\text{reas}}^{(\ell)}$ enter this dominating subset. Detail in Appx. B.

Based on this insight, we introduce two global gains $g_{\text{perc}} \geq 1$ and $g_{\text{reas}} \geq 1$ for perception and reasoning heads, respectively, while leaving all other heads neutral with a factor of 1. Formally, for each layer $\ell$ and head $h$, we define

$$g^{(h,\ell)} = g_{\text{perc}} \cdot \mathbf{1}\Big[h \in \mathcal{H}_{\text{perc}}^{(\ell)}\Big] + g_{\text{reas}} \cdot \mathbf{1}\Big[h \in \mathcal{H}_{\text{reas}}^{(\ell)}\Big] + 1 \cdot \mathbf{1}\Big[h \notin \mathcal{H}_{\text{perc}}^{(\ell)} \cup \mathcal{H}_{\text{reas}}^{(\ell)}\Big], \quad (6)$$

where $\mathbf{1}[\cdot]$ is the *indicator function*, which returns 1 if true and 0 otherwise. Thus, $g^{(h,\ell)}$ equals $g_{\mathrm{perc}}$ for perception heads, $g_{\mathrm{reas}}$ for reasoning heads, and 1 for all other heads. This compact formulation makes explicit that each head is either amplified according to its functional role or left unchanged.

For each layer $\ell$, we apply gains after per-head outputs are computed but before the output projection. Let $Y^{(\ell)} = \mathrm{Concat}(O^{(1,\ell)}, \ldots, O^{(H,\ell)})\, W_O^{(\ell)}$ be the standard output, the rescaled output is:

$$Y_{\mathrm{out}}^{(\ell)} = \mathrm{Concat}\big(g^{(1,\ell)}O^{(1,\ell)}, \ldots, g^{(H,\ell)}O^{(H,\ell)}\big)\, W_O^{(\ell)}. \qquad (7)$$

Conceptually, class-conditioned rescaling enhances perception heads in shallow layers and reasoning heads in deeper layers, thereby addressing the two major failure modes: reinforcing visual grounding to mitigate *Perceptual Bias*, and strengthening logical consistency to reduce *Reasoning Drift*.

Crucially, because the rescaled output $Y_{\mathrm{out}}^{(\ell)}$ is injected into the residual stream and normalized before passing forward, these adjustments do not remain local. Their influence compounds across subsequent layers, meaning that amplifying the correct functional heads shapes the model's global reasoning trajectory. In this way, class-conditioned rescaling provides a lightweight yet interpretable mechanism to strengthen the macro-level functions most critical for reducing hallucination.

The design is intentionally minimal: both class gains are *amplifying* and *layer-agnostic*, aiming to provide a lightweight and implementation-friendly means to emphasise screened heads without altering the underlying attention weights or value projections.

## 4 EXPERIMENTS

In this section, we conduct experiments to address the following research questions:

❖ **RQ1:** Can the proposed method consistently reduce hallucinations across multimodal reasoning tasks and benchmarks, demonstrating gains over baselines? ❖ **RQ2:** How do the amplification of perception and reasoning heads contribute to overall effectiveness? ❖ **RQ3:** How do different configurations of layer boundaries ($\ell_{\mathrm{perc}}$ and $\ell_{\mathrm{reas}}$) affect the overall performance of the model? ❖ **RQ4:** How do the visual-attention ratio thresholds ($\tau_{\mathrm{perc}}$ and $\tau_{\mathrm{reas}}$) impact the head screening process and the method's overall performance?

**Environment.** Experiments with `Ocean-R1` use a 24-core Xeon CPU and an A40 GPU, while `R1-Onevision` and `Kimi-VL` use a 16-core Xeon CPU and an A800 GPU. Details in Appx. D.

**Datasets.** We evaluate on **6 benchmarks**: ❶ **Mathematics Reasoning**: MathVista$_{\mathrm{mini}}$ (Lu et al., 2023), MathVision$_{\mathrm{mini}}$ (Wang et al., 2024b); ❷ **Visual Reasoning**: CLEVR (Johnson et al., 2017), HallusionBench (Guan et al., 2024); ❸ **Multimodal Integration**: MMStar (Chen et al., 2024), SEED-Bench (Li et al., 2024). Details in Appx. E.

**Baselines.** We compare with **3 inference-time baselines**: VCD (Leng et al., 2024) contrasts original vs. counterfactual image views; AGLA (An et al., 2025) fuses global and local views to improve grounding; CGD (Deng et al., 2024) applies CLIP-guided priors during decoding. Details in Appx. F.

### 4.1 EFFECTIVENESS (RQ1)

To evaluate the effectiveness of our proposed plugin for MLRMs, we compare it against three state-of-the-art hallucination mitigation baselines as well as the vanilla backbone models. The results, summarized in Table 1, cover six benchmarks spanning mathematics reasoning, visual reasoning, and multimodal integration. In *Obs. 1* and *Obs. 2*, we further analyze accuracy and efficiency, confirming that our *plug-and-play* design delivers performance gains with *near-zero* inference overhead.

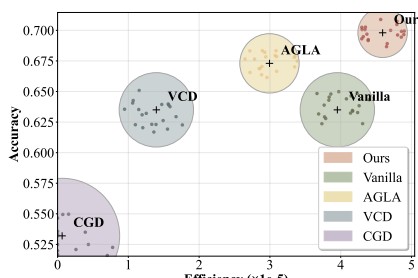

Figure 3: Efficiency comparison. Our method achieves the best efficiency while simultaneously improving accuracy. The x-axis reports $(\mathrm{Acc}/\mathrm{AvgBatchTime})^2$ computed over 200 samples from HallusionBench with `Kimi-VL`.

❖ *Obs. 1:* **Jointly optimizing perception and reasoning yields broad improvements.** Our method achieves best performance on nearly 95% of the tasks, with an average gain of 4.8% over the vanilla baseline. Unlike prior approaches, which often trade improvements between reasoning (mathematics) and perception (visual)

Table 1: **Overall results.** Performance across six benchmarks covering mathematics, visual reasoning, and multimodal integration. **Bold** indicate the best results, and underlined numbers indicate the second best.

| Method | Model | Mathematics Reasoning | | Visual Reasoning | | | Multimodal Integration | | | |
| | | MathVista$_{mini}$ | MathVision$_{mini}$ | HallusionBench | | CLEVR | MMStar | | SEED-Bench | |
| | | Acc | Acc | Acc | F1 | Acc | Acc | F1 | Acc | F1 |
|---|---|---|---|---|---|---|---|---|---|---|
| Vanilla | Kimi-VL A3B-Thinking | $63.48_{\pm0.91}$ | $56.24_{\pm1.42}$ | $64.76_{\pm0.52}$ | $64.97_{\pm0.98}$ | $76.77_{\pm1.03}$ | $59.76_{\pm0.49}$ | $42.75_{\pm1.08}$ | $66.26_{\pm0.41}$ | $49.54_{\pm1.12}$ |
| VCD | | $63.51_{\pm0.82}$ | $56.14_{\pm0.79}$ | $\underline{65.68}_{\pm0.62}$ | $\underline{67.91}_{\pm0.54}$ | $\underline{78.01}_{\pm1.23}$ | $59.48_{\pm0.61}$ | $42.61_{\pm1.12}$ | $66.52_{\pm1.53}$ | $49.69_{\pm0.82}$ |
| CGD | | $53.18_{\pm1.55}$ | $49.3_{\pm0.80}$ | $52.13_{\pm0.81}$ | $55.73_{\pm1.06}$ | $58.23_{\pm1.33}$ | $48.73_{\pm1.15}$ | $31.09_{\pm0.75}$ | $43.98_{\pm1.59}$ | $34.88_{\pm1.38}$ |
| AGLA | | $\underline{67.32}_{\pm1.11}$ | $\underline{58.88}_{\pm1.47}$ | $61.36_{\pm1.11}$ | $63.51_{\pm1.23}$ | $71.73_{\pm1.40}$ | $\underline{63.76}_{\pm0.85}$ | $\underline{47.06}_{\pm1.01}$ | $\underline{69.27}_{\pm1.39}$ | $\underline{51.48}_{\pm0.99}$ |
| Ours | | $\mathbf{69.78}_{\pm0.82}$ | $\mathbf{60.54}_{\pm0.98}$ | $\mathbf{68.19}_{\pm1.07}$ | $\mathbf{68.32}_{\pm0.80}$ | $\mathbf{81.73}_{\pm1.30}$ | $\mathbf{66.49}_{\pm1.32}$ | $\mathbf{49.78}_{\pm0.63}$ | $\mathbf{69.74}_{\pm1.18}$ | $\mathbf{53.62}_{\pm1.22}$ |
| Vanilla | Ocean-R1 7B-Instruct | $54.58_{\pm0.87}$ | $20.05_{\pm0.80}$ | $49.41_{\pm0.61}$ | $49.45_{\pm1.60}$ | $38.26_{\pm1.50}$ | $45.24_{\pm0.41}$ | $29.38_{\pm1.12}$ | $59.76_{\pm0.92}$ | $42.61_{\pm0.96}$ |
| VCD | | $54.51_{\pm1.44}$ | $19.92_{\pm1.34}$ | $\underline{50.57}_{\pm0.52}$ | $\underline{50.66}_{\pm0.88}$ | $\underline{39.52}_{\pm1.21}$ | $45.48_{\pm0.96}$ | $29.23_{\pm0.43}$ | $59.98_{\pm1.30}$ | $42.87_{\pm1.01}$ |
| CGD | | $46.81_{\pm0.88}$ | $10.4_{\pm0.96}$ | $33.72_{\pm0.56}$ | $33.34_{\pm1.06}$ | $20.77_{\pm1.43}$ | $36.24_{\pm1.45}$ | $21.51_{\pm1.25}$ | $37.76_{\pm0.59}$ | $17.55_{\pm1.29}$ |
| AGLA | | $\underline{57.49}_{\pm1.09}$ | $\underline{23.69}_{\pm0.73}$ | $45.2_{\pm0.92}$ | $45.07_{\pm0.78}$ | $33.51_{\pm1.29}$ | $\underline{48.48}_{\pm0.53}$ | $\underline{32.27}_{\pm0.76}$ | $\underline{63.01}_{\pm1.04}$ | $\underline{45.95}_{\pm0.61}$ |
| Ours | | $\mathbf{59.32}_{\pm0.88}$ | $\mathbf{26.01}_{\pm0.50}$ | $\mathbf{53.64}_{\pm1.58}$ | $\mathbf{53.77}_{\pm0.49}$ | $\mathbf{43.01}_{\pm1.50}$ | $\mathbf{50.77}_{\pm1.14}$ | $\mathbf{34.11}_{\pm0.99}$ | $\mathbf{66.51}_{\pm1.07}$ | $\mathbf{49.66}_{\pm1.33}$ |
| Vanilla | R1-Onevision 7B | $59.92_{\pm1.13}$ | $33.54_{\pm0.72}$ | $58.26_{\pm1.59}$ | $58.49_{\pm1.55}$ | $47.98_{\pm1.00}$ | $56.26_{\pm0.53}$ | $39.11_{\pm0.61}$ | $68.48_{\pm1.53}$ | $52.06_{\pm0.67}$ |
| VCD | | $59.69_{\pm1.34}$ | $33.44_{\pm1.37}$ | $\underline{58.96}_{\pm0.53}$ | $\underline{59.03}_{\pm0.54}$ | $\underline{52.24}_{\pm0.90}$ | $\underline{56.74}_{\pm0.62}$ | $\underline{39.61}_{\pm1.25}$ | $68.76_{\pm0.78}$ | $\underline{52.31}_{\pm1.36}$ |
| CGD | | $52.58_{\pm1.56}$ | $28.12_{\pm1.19}$ | $48.89_{\pm1.47}$ | $48.82_{\pm1.16}$ | $34.73_{\pm1.48}$ | $43.48_{\pm1.28}$ | $26.87_{\pm1.53}$ | $63.23_{\pm0.60}$ | $46.58_{\pm0.87}$ |
| AGLA | | $\mathbf{60.21}_{\pm1.48}$ | $\underline{37.03}_{\pm1.58}$ | $55.69_{\pm0.76}$ | $55.75_{\pm0.44}$ | $50.51_{\pm1.41}$ | $54.49_{\pm0.49}$ | $37.86_{\pm1.42}$ | $68.11_{\pm1.17}$ | $51.65_{\pm1.48}$ |
| Ours | | $\underline{60.09}_{\pm1.48}$ | $\mathbf{39.12}_{\pm0.44}$ | $\mathbf{60.77}_{\pm0.98}$ | $\mathbf{60.88}_{\pm0.54}$ | $\mathbf{62.77}_{\pm1.02}$ | $\mathbf{58.02}_{\pm0.83}$ | $\mathbf{40.94}_{\pm0.49}$ | $\mathbf{69.52}_{\pm1.52}$ | $\mathbf{53.01}_{\pm1.46}$ |

tasks, our patch consistently enhances both sides. For example, VCD yields clear gains on visual benchmarks such as HallusionBench but shows limited benefits on mathematics reasoning tasks, whereas our method provides more balanced improvements across both categories. Cases in Appx. H.

❖ *Obs. 2:* **Dual-stage gains come with negligible efficiency cost.** As illustrated in Figure 3, our patch introduces only marginal overhead: on HallusionBench (200 samples), vanilla models take ∼101s on average, while our method adds merely ∼2s (103s total). In contrast, baseline methods (VCD, CGD, AGLA) incur $1.2\times - 6.6\times$ total inference time. This makes our approach highly pluggable and cost-efficient, substantially improving reliability without sacrificing deployability. A theoretical analysis and more results are provided in Appx. C.

### 4.2 ABLATION ON HEAD AMPLIFICATION (RQ2)

To isolate the effect of rescaling each functional group, we conduct ablations as summarized in Table 2. Concretely, we evaluate: (i) **w/o reasoning**, which identifies and enhances only *perception* heads; and (ii) **w/o perception**, which identifies and enhances only *reasoning* heads. All other hyperparameters are kept identical to the full plugin to ensure a fair comparison.

❖ *Obs. 3:* **Perception-only vs. reasoning-only effects are task-skewed and non-additive.** In most settings, **w/o reasoning** (enhancing only perception heads) impacts visual tasks more, while **w/o perception** (enhancing only reasoning heads) drives the gains in mathematics. However, we also observe counterexamples: on R1-OneVision with MathVision$_{mini}$, strengthening a single group alone leads to a $-3.91\%$ drop, whereas combining both groups yields a $+5.58\%$ gain. This aligns with our claim in Sec. 2.3 that hallucination is not a single-capability failure but emerges from the *interaction* between perception and reasoning stages.

❖ *Obs. 4:* **Model-specific reliance on perception vs. reasoning.** On multimodal integration tasks, we observe heterogeneous behaviors across architectures. For instance, on Kimi-VL with MMStar, **w/o reasoning** yields an $+6.71\%$ improvement, whereas on Ocean-R1 the same setting causes a $-1.51\%$ drop. This suggests that different MLRM architectures may emphasize distinct functional capacities, highlighting the importance of model-specific balancing between perception and reasoning.

### 4.3 IMPACT OF BOUNDARY CONFIGURATIONS (RQ3)

Our plugin first identifies perception and reasoning heads, followed by rescaling. A key factor is the choice of layer boundaries $(\ell_{\mathrm{perc}}, \ell_{\mathrm{reas}})$, which determines which heads are classified as perception- or reasoning-oriented. Figure 4 reports results on three representative datasets, each drawn from one of the evaluation categories, showing how performance varies under different boundary configurations. Based on these results, we adopt $\ell_{\mathrm{perc}} = 7$ and $\ell_{\mathrm{reas}} = 3$ as our final configuration.

Table 2: **Ablation of perception- and reasoning-head rescaling.** We compare the full plugin with variants that disable either perception or reasoning rescaling, alongside the vanilla model.

| Method | Model | Mathematics Reasoning | | Visual Reasoning | | | Multimodal Integration | | | |
| --- | --- | --- | --- | --- | --- | --- | --- | --- | --- | --- |
| | | MathVista$_{mini}$ | MathVision$_{mini}$ | HallusionBench | | CLEVR | MMStar | | SEED-Bench | |
| | | Acc | Acc | Acc | F1 | Acc | Acc | F1 | Acc | F1 |
| Vanilla | Kimi-VL A3B-Thinking | 63.48$_{\pm0.91}$ | 56.24$_{\pm1.42}$ | 64.76$_{\pm0.52}$ | 64.97$_{\pm0.98}$ | 76.77$_{\pm1.03}$ | 59.76$_{\pm0.49}$ | 42.75$_{\pm1.08}$ | 66.26$_{\pm0.41}$ | 49.54$_{\pm1.12}$ |
| w/o Reason | | 69.2$_{\pm0.59}$ | 58.54$_{\pm0.77}$ | 65.78$_{\pm0.90}$ | 65.94$_{\pm1.15}$ | 82.27$_{\pm0.49}$ | 66.47$_{\pm1.44}$ | 49.76$_{\pm0.58}$ | 68.49$_{\pm1.46}$ | 52.45$_{\pm0.78}$ |
| w/o Percept | | 69.37$_{\pm0.62}$ | 62.84$_{\pm0.61}$ | 65.32$_{\pm0.42}$ | 65.49$_{\pm0.66}$ | 81.48$_{\pm0.89}$ | 63.23$_{\pm0.78}$ | 46.31$_{\pm0.61}$ | 69.49$_{\pm1.14}$ | 53.62$_{\pm0.85}$ |
| Ours | | 69.78$_{\pm0.82}$ | 60.54$_{\pm0.98}$ | 68.19$_{\pm1.07}$ | 68.32$_{\pm0.80}$ | 81.73$_{\pm1.30}$ | 66.49$_{\pm1.32}$ | 49.78$_{\pm0.63}$ | 69.74$_{\pm1.18}$ | 53.64$_{\pm1.22}$ |
| Vanilla | Ocean-R1 7B-Instruct | 54.58$_{\pm0.87}$ | 20.05$_{\pm0.80}$ | 49.41$_{\pm0.61}$ | 49.45$_{\pm1.60}$ | 38.26$_{\pm1.50}$ | 45.24$_{\pm0.41}$ | 29.38$_{\pm1.12}$ | 59.76$_{\pm0.92}$ | 42.61$_{\pm0.96}$ |
| w/o Reason | | 55.11$_{\pm0.71}$ | 26.01$_{\pm0.58}$ | 48.15$_{\pm1.14}$ | 48.16$_{\pm1.55}$ | 40.01$_{\pm0.86}$ | 43.73$_{\pm1.51}$ | 28.18$_{\pm1.43}$ | 60.52$_{\pm0.70}$ | 43.31$_{\pm0.75}$ |
| w/o Percept | | 58.22$_{\pm0.66}$ | 24.35$_{\pm1.60}$ | 54.37$_{\pm1.34}$ | 54.47$_{\pm0.67}$ | 41.01$_{\pm0.82}$ | 50.48$_{\pm0.73}$ | 33.93$_{\pm1.15}$ | 61.99$_{\pm0.98}$ | 44.83$_{\pm1.26}$ |
| Ours | | 59.32$_{\pm0.88}$ | 25.68$_{\pm0.50}$ | 53.64$_{\pm1.58}$ | 53.77$_{\pm0.49}$ | 43.01$_{\pm1.50}$ | 50.77$_{\pm1.14}$ | 34.11$_{\pm0.99}$ | 66.51$_{\pm1.07}$ | 49.66$_{\pm1.33}$ |
| Vanilla | R1-Onevision 7B | 59.92$_{\pm1.13}$ | 33.54$_{\pm0.72}$ | 58.26$_{\pm1.59}$ | 58.49$_{\pm1.55}$ | 47.98$_{\pm1.00}$ | 56.26$_{\pm0.53}$ | 39.11$_{\pm0.61}$ | 68.48$_{\pm1.53}$ | 52.06$_{\pm0.67}$ |
| w/o Reason | | 59.19$_{\pm1.18}$ | 33.9$_{\pm1.53}$ | 60.66$_{\pm0.62}$ | 60.83$_{\pm1.23}$ | 48.77$_{\pm0.85}$ | 57.77$_{\pm1.07}$ | 40.67$_{\pm1.38}$ | 71.23$_{\pm0.50}$ | 55.41$_{\pm1.30}$ |
| w/o Percept | | 59.29$_{\pm1.38}$ | 29.63$_{\pm1.12}$ | 59.42$_{\pm0.70}$ | 59.51$_{\pm0.64}$ | 55.99$_{\pm0.42}$ | 53.73$_{\pm0.48}$ | 36.82$_{\pm1.47}$ | 68.51$_{\pm0.93}$ | 52.21$_{\pm0.71}$ |
| Ours | | 60.09$_{\pm1.48}$ | 39.12$_{\pm0.44}$ | 60.77$_{\pm0.98}$ | 60.88$_{\pm0.54}$ | 62.77$_{\pm1.02}$ | 58.02$_{\pm0.83}$ | 40.94$_{\pm0.49}$ | 69.52$_{\pm1.52}$ | 53.01$_{\pm1.46}$ |

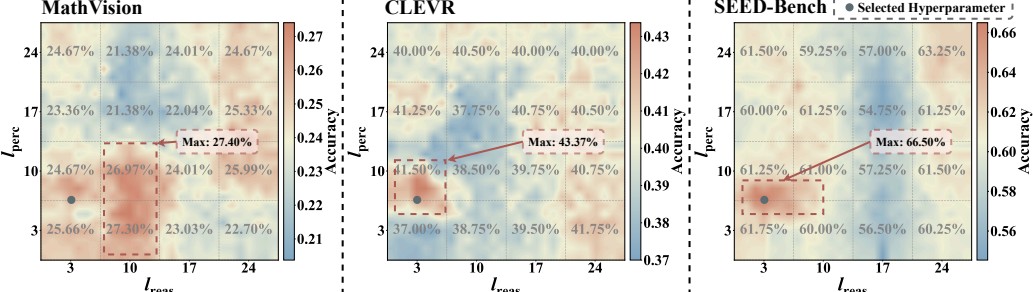

Figure 4: **Boundary sweep on `Ocean-R1`.** We fix four hyperparameters ($g_{\text{reas}}=1.3$, $\tau_{\text{reas}}=0.01$, $g_{\text{perc}}=1.16$, $\tau_{\text{perc}}=0.22$) and vary the layer boundaries to assess performance sensitivity.

❖ *Obs. 5:* **Task-dependent boundary bands rather than a single split.** For *visual-dominant* tasks (CLEVR, SEED-Bench), the best scores concentrate in the lower-left of the boundary grid, pointing to $\ell_{\text{perc}} \approx 5$–9 and $\ell_{\text{reas}} \approx 2$–5. Operationally, layers 0–5 act as perception, and $\gtrsim 9$ as reasoning. By contrast, the *reasoning-dominant* task (MathVision) peaks when $\ell_{\text{reas}} \approx 10$, forming a high-performance region that *does not overlap* with the visual-task region. We hypothesize that this band marks a *perception-to-reasoning transition zone*, where control shifts from visual evidence aggregation to symbolic inference. Hence, the boundaries behave like *task-dependent bands*, with a mid-depth transition zone rather than a single crisp threshold.

❖ *Obs. 6:* **Deep-layer reasoning exists but is interaction-sensitive.** Strengthening reasoning at deeper layers (e.g., $\ell_{\text{reas}} \approx 24$; right-side region) also yields strong performance, suggesting the presence of reasoning-related functional modules. However, their effect is coupled with perception ($\ell_{\text{perc}}$) and task type, inducing $\sim 5\%$ performance swings across settings. This supports our choice to keep $\ell_{\text{perc}}$ and $\ell_{\text{reas}}$ in shallow ($\leq 10$) layers for more consistent performance across diverse tasks. Interestingly, the low Acc observed in the intermediate range ($\ell \in [10, 17]$), when contrasted with both shallow and deep high-Acc regions, reveals the existence of a transitional zone where perception and reasoning functions are more intricately intertwined rather than cleanly separated.

### 4.4 EFFECT OF ATTENTION RATIO THRESHOLDS (RQ4)

We further investigate the influence of multiplicative gains ($g_{\text{reas}}, g_{\text{perc}}$) and ratio thresholds ($\tau_{\text{reas}}, \tau_{\text{perc}}$) on model performance. As shown in Figure 5, we report results under the same evaluation setup as Table 1, with default hyperparameters specified in Figure 5.

❖ *Obs. 7:* **Multiplicative gains show task-dependent yet partially transferable effects.** Peaks appear at similar values (e.g., $g_{\text{reas}} = 1.30$, $g_{\text{perc}} = 1.16$) for MathVista$_{mini}$ and MMStar, yielding $\sim 15\%$ and $\sim 6\%$ gains, respectively, and suggesting some cross-task generality. From the overall trend, $g_{\text{reas}}$ shows a steady pattern, delivering $\sim 10\%$ improvement already at 1.10, whereas $g_{\text{perc}}$ fluctuates more noticeably across datasets and settings.

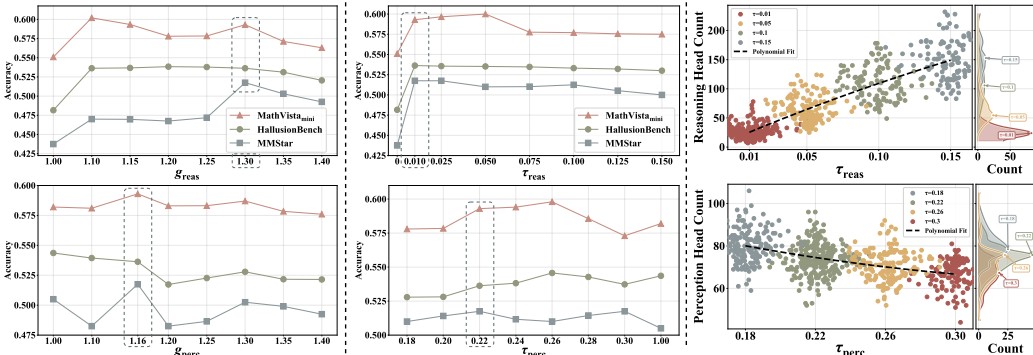

Figure 5: **Analysis of multiplicative gains and ratio thresholds.** On `Ocean-R1` with fixed boundaries $\ell_{\text{reas}}$=3, $\ell_{\text{perc}}$=7: *Left*—performance on three datasets under varying multiplicative gains $(g_{\text{reas}}, g_{\text{perc}})$; *Middle*—impact of ratio thresholds $(\tau_{\text{reas}}, \tau_{\text{perc}})$ on performance; *Right*—number of identified heads as a function of $(\tau_{\text{reas}}, \tau_{\text{perc}})$ (small horizontal jitters are added for visual clarity). Default settings are $g_{\text{reas}}$=1.30, $\tau_{\text{reas}}$=0.01, $g_{\text{perc}}$=1.16, $\tau_{\text{perc}}$=0.22.

❖ *Obs. 8:* **Ratio thresholds control intervention sparsity and quality.** As shown in the right panel of Figure 5, varying the ratio thresholds directly changes the number of intervened heads (with token-dependent variability). For $\tau_{\text{reas}}$, strong performance concentrates when $\sim 50$–$150$ heads are selected on average (about $6.4\%$ of all heads), indicating a *sparse* intervention. As the target set grows (approximately $6\% \rightarrow 18\%$), performance degrades gradually (e.g., error $0.59 \rightarrow 5.7$), consistent with dilution from less functional heads.

## 5 RELATED WORK

**Interpretability of Multimodal Reasoning.** Long-chain reasoning is critical for complex multimodal tasks, yet the opacity of current models limits transparency. Existing research follows two main directions. Prompting- and data-driven methods extend Chain-of-Thought reasoning to visual inputs (Wei et al., 2022), either through functionally distinct prompts (Zheng et al., 2023) or curated datasets with annotated reasoning traces and visual highlights (Shao et al., 2024). Architectural and pipeline designs restructure the reasoning process via modular agents (Chen et al., 2023b) or explanation-driven preference optimization (Zhang et al., 2024b). While these approaches expose latent rationales, they often demand heavy supervision or handcrafted prompting. A growing consensus in this line of work is that multimodal reasoning unfolds in stages of perception and symbolic reasoning, which motivates the stage-aware perspective underlying our method (Zheng et al., 2023; Shao et al., 2024).

**Multimodal Hallucination.** Prior studies attribute hallucination in MLRMs to three recurring causes: **(i)** strong language priors and co-occurrence biases, **(ii)** insufficient cross-modal alignment, and **(iii)** training or decoding artifacts (Yue et al., 2024). Analyses across benchmarks such as CHAIR (Rohrbach et al., 2018), POPE (Li et al., 2023b), and MMHal-Bench (Sun et al., 2024) consistently support these views. Mitigation strategies fall into three families. Contrastive decoding perturbs inputs or prompts to down-weight unreliable tokens (e.g., VCD (Leng et al., 2024), ICD (Wang et al., 2024c), LCD (Manevich & Tsarfaty, 2024)). Alignment and preference learning strengthen visual grounding through DPO (Yang et al., 2025b) variants or hallucination-aware training (e.g., HACL (Jiang et al., 2024)). External validation and tool grounding enforce consistency using detectors or CLIP-based guidance (e.g., MARINE (Zhao et al., 2025b)).

## 6 CONCLUSION

We introduce a lightweight, interpretable plugin for MLRMs that identifies perception- and reasoning-oriented heads and selectively enhances them under a minimal-editing principle. Unlike prior decoding-time or training-based approaches, our method is *plug-and-play*, requiring no retraining or architectural changes. Furthermore, the method improves accuracy without introducing any discernible *inference-time overhead*. Across diverse benchmarks, it delivers balanced gains on perception-heavy and reasoning-heavy tasks, confirming that targeted head control can mitigate hallucination without compromising efficiency. We believe this drop-in, cost-free intervention offers a practical path to more reliable and interpretable multimodal reasoning in the deployment stage.

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

## THE USE OF LARGE LANGUAGE MODELS (LLMs)

In preparing this manuscript, we used a large language model (LLM) exclusively for stylistic refinement, such as polishing grammar, improving clarity, and enhancing readability. The LLM was not involved in formulating research questions, designing methods, analyzing data, or drawing conclusions. All scientific content, ideas, and experimental results are entirely the work of the authors. The LLM served purely as an auxiliary tool for language editing.

## LIMITATION, DISCUSSION, AND FUTURE WORK

**Beyond a single boundary.** Our experiments in Sec. 4.3 and Sec. 4.4 show that the division between perception and reasoning is not determined by a crisp threshold. Instead, performance peaks appear in *bands* of layers, and the optimal $\ell_{\text{perc}}$ and $\ell_{\text{reas}}$ vary across tasks. This suggests that different layers may play mixed roles, and that the model undergoes a gradual transition from perception to reasoning rather than a clean separation. Future work could build on interpretability studies to map these transitions more precisely, moving from a one-dimensional boundary to richer structural patterns.

**Toward more adaptive selection.** Our method follows the *minimal editing* principle, amplifying a small set of heads without attenuating others. Formally, the gain in task alignment is $\Delta = \sum_{h \in \mathcal{H}} (\gamma_h - 1) u_h$, where $u_h$ measures the usefulness of head $h$. In this work, we identify perception and reasoning heads using simple ratio thresholds. A natural extension is to design more fine-grained classifiers that score each head by multiple signals (e.g., depth, modality ratio, consistency). An adaptive scheme could then select $S(x)$ depending on the input $x$, and set $\gamma_h > 1$ only for those heads. While this may yield larger $\Delta$, it also risks higher variance and efficiency costs. Thus, the challenge is to balance stronger, adaptive selection with the stability and efficiency of the current plug-and-play design.

## A   NOTATIONS AND DEFINITIONS

Table 3: Notations and Definitions

| Notation | Definition |
|---|---|
| $\mathcal{T}_v, \mathcal{T}_t, \mathcal{T}_q$ | Sets of vision, text tokens and query tokens, respectively. |
| $l, h$ | Indices for layer and attention head in a Transformer, respectively. |
| $X^{(l)}$ | Input hidden states at layer $l$. |
| $W_Q^{(h,l)}, W_K^{(h,l)}, W_V^{(h,l)}$ | Query, key, and value projection matrices for head $h$ in layer $l$. |
| $A^{(h,l)}$ | Attention matrix for head $h$ in layer $l$, $A^{(h,l)} \in \mathbb{R}^{N \times N}$. |
| $a_{ij}^{(h,l)}$ | Attention weight from query token $i$ to key token $j$ in matrix $A^{(h,l)}$. |
| $O^{(h,l)}$ | Output vector for head $h$ in layer $l$, $O_h^{(l)} \in \mathbb{R}^{d_{model}}$. |
| $V^{(h,l)}$ | Value tensor for head $h$ in layer $l$. |
| $Y^{(l)}$ | Standard multi-head output of the Transformer block at layer $l$. |
| $\mathcal{L}_{perc}, \mathcal{L}_{reas}$ | Sets of perception layers and reasoning layers, respectively. |
| $l_{perc}, l_{reas}$ | Boundaries for the last layer of the perception stage and the first layer of the reasoning stage. |
| $S_v^{(l)}(h), S_t^{(l)}(h)$ | Average attention ratio allocated to visual/textual tokens by head $h$ in layer $l$. |
| $\tau_{perc}, \tau_{reas}$ | Thresholds of visual attention ratio to distinguish perception and reasoning heads. |

| Table 3 – Continued from previous page | |
|---|---|
| **Notation** | **Definition** |
| $\mathcal{H}$ | The full set of attention heads in a given multi-head attention (MHA) sub-layer. |
| $\mathcal{H}_{perc}^{(l)}, \mathcal{H}_{reas}^{(l)}$ | Sets of perception-oriented and reasoning-oriented heads identified in layer $l$. |
| $g_{perc}, g_{reas}$ | Global gain factors applied to perception and reasoning heads, respectively. |
| $\alpha_{perc}, \alpha_{reas}$ | Scalar gain assigned to a head based on its attention ratio (used interchangeably with $g_{perc}, g_{reas}$). |
| $g^{(h,l)}$ | Specific gain value applied to head $h$ in layer $l$. |
| $O_{\mathcal{S}}$ | Aggregated output of a subset of heads $\mathcal{S} \subseteq \mathcal{H}$. |

## B  SELECTIVE ENHANCEMENT AS OUR POLICY CHOICE

**Problem and Objective (Minimal Editing Principle).**  When intervening on attention heads, our objective is to *maximize the correction of hallucination* while *minimizing unnecessary edits* to the model's internal representations. We formalize this idea as the **Minimal Editing Principle**: an effective intervention should (1) amplify signals that are already verified as beneficial, and (2) avoid suppressing or perturbing heads whose functions remain uncertain.

Let $\mathcal{H}$ denote the full set of attention heads in a given multi-head attention (MHA) sub-layer. Each head $h \in \mathcal{H}$ produces an output vector $\mathbf{O}_h^{(l)} \in \mathbb{R}^{d_{\text{model}}}$, and the aggregated MHA output can be expressed as

$$\tilde{Y}^{(l)} = \sum_{h \in \mathcal{H}} \gamma_h^{(l)} \mathbf{O}_h^{(l)}, \tag{8}$$

where $\gamma_h^{(l)}$ is a multiplicative gain applied to head $h$. In this formulation, any intervention corresponds to editing the gain pattern $\{\gamma_h^{(l)}\}_{h \in \mathcal{H}}$. *Minimal editing* requires that such modifications keep the original contributions of most heads, while selectively reinforcing those aligned with perception or reasoning.

Intuitively, this principle reflects a conservative stance. We cannot guarantee that non-target heads are harmful. If they are attenuated indiscriminately, the model may lose useful functions and suffer collateral damage. The safer strategy is therefore to intervene as little as possible: amplify only the identified functional heads, while keeping the rest unchanged to preserve the existing representation.

**Formal Setup and Minimal Assumptions.**  To make analysis tractable and verifiable, we adopt two minimal assumptions: (1) interventions operate *after attention computation but before residual and normalization*, ensuring that changes are linear and directly comparable; (2) all enhanced heads share a common amplification factor $\alpha > 1$, while all attenuated heads (if any) share a common factor $\beta < 1$.

These assumptions preserve the generality of our formulation while guaranteeing that differences between strategies can be derived in closed form. In the next subsection, we instantiate four policies under this setup and analyze their functional differences.

**Four Strategies and a Core Proposition.**  Under the above setup, we distinguish four representative policies for modifying head gains:

- **Strategy A (Selective Enhancement).**

$$\gamma_h^{(l)} = \begin{cases} \alpha, & h \in \mathcal{H}_{\text{enhance}}, \\ 1, & h \in \mathcal{H} \setminus \mathcal{H}_{\text{enhance}}. \end{cases} \tag{9}$$

- **Strategy B (Selective Attenuation).**

$$\gamma_h^{(l)} = \begin{cases} \beta, & h \in \mathcal{H}_{\text{attenuate}}, \\ 1, & h \in \mathcal{H} \setminus \mathcal{H}_{\text{attenuate}}. \end{cases} \tag{10}$$

- **Strategy C (Bipolar Scaling).**

$$\gamma_h^{(l)} = \begin{cases} \alpha, & h \in \mathcal{H}_{\text{enhance}}, \\ \beta, & h \in \mathcal{H}_{\text{attenuate}}. \end{cases} \tag{11}$$

- **Strategy D (Mixed Policy).**

$$\gamma_h^{(l)} = \begin{cases} \alpha, & h \in \mathcal{H}_{\text{enhance}}, \\ \beta, & h \in \mathcal{H}_{\text{attenuate}}, \\ 1, & h \in \mathcal{H}_{\text{neutral}}. \end{cases} \tag{12}$$

**Proposition (Difference Equation).** Let $\mathbf{O}_{\mathcal{S}} = \sum_{h \in \mathcal{S}} \mathbf{O}_h^{(l)}$ denote the aggregated output of a subset of heads $\mathcal{S} \subseteq \mathcal{H}$. Then, relative to Strategy A, any scheme that introduces attenuation (C or D) differs by an explicit subtractive term:

$$\tilde{Y}_C^{(l)} - \tilde{Y}_A^{(l)} = (\beta - 1)\mathbf{O}_{\mathcal{H}_{\text{att}}}, \tag{13}$$

$$\tilde{Y}_D^{(l)} - \tilde{Y}_A^{(l)} = (\beta - 1)\mathbf{O}_{\mathcal{H}_{\text{att}}}, \tag{14}$$

where $\mathcal{H}_{\text{att}}$ denotes the set of attenuated heads.

*Proof.* Both Strategy A and Strategy C/D share the amplified component $\alpha\,\mathbf{O}_{\mathcal{H}_{\text{enh}}}$. The difference arises in the treatment of non-enhanced heads: Strategy A preserves their outputs unchanged, while C and D rescale them by $\beta < 1$. This introduces a correction term $(\beta - 1)\mathbf{O}_{\mathcal{H}_{\text{att}}}$, representing the explicit removal of information contributed by attenuated heads.

**Interpretation.** This difference equation makes the design trade-off explicit: ❖ Strategy A amplifies useful heads while keeping all others intact. ❖ Strategies C and D apply an additional subtraction to all attenuated heads, implicitly assuming they are harmful. Since non-target heads may still carry latent but beneficial functions, attenuation risks collateral degradation. Thus, the *minimal editing* choice is Strategy A, which strengthens the identified heads without disturbing the rest of the model.

**Expectation-Level Intuition: Why Selective Enhancement is Safer.** Consider a task-aligned direction $v \in \mathbb{R}^{d_{\text{model}}}$, and let each head's contribution be

$$u_h := \langle \mathbf{O}_h^{(l)}, v \rangle, \tag{15}$$

which measures how much the head $h$ aligns with the task. For any headset $\mathcal{S}$, we then define its total contribution as

$$U(\mathcal{S}) = \sum_{h \in \mathcal{S}} u_h. \tag{16}$$

Ignoring LayerNorm rescaling, the alignment change caused by a gain pattern $\{\gamma_h^{(l)}\}$ is

$$\Delta(\{\gamma\}) \approx \sum_{h \in \mathcal{H}} (\gamma_h^{(l)} - 1)\,u_h, \tag{17}$$

which simply adds the weighted shifts of each head relative to the neutral baseline ($\gamma_h = 1$).

**Three Strategies.** Applying this formulation, the alignment increments become:

$$\Delta_A \approx (\alpha - 1)\,U(\mathcal{H}_{\text{enhance}}), \tag{18}$$

corresponding to *selective enhancement*, where only useful heads are amplified.

$$\Delta_B \approx (\beta - 1)\,U(\mathcal{H}_{\text{attenuate}}), \tag{19}$$

for *selective attenuation*, where some heads are suppressed while all others remain neutral.

$$\Delta_C \approx (\alpha - 1)\,U(\mathcal{H}_{\text{enhance}}) + (\beta - 1)\,U(\mathcal{H}_{\text{attenuate}}), \tag{20}$$

for *bipolar scaling*, where enhancement and attenuation are applied simultaneously.

**Assumption (non-harmful heads).** Prior interpretability studies indicate that most attention heads carry diverse or weakly positive roles rather than being systematically harmful (Olsson et al., 2022). Formally, this suggests

$$\mathbb{E}[\,U(\mathcal{H}_{\text{attenuate}})\,] \;\geq\; 0, \tag{21}$$

meaning that, on average, attenuated heads are not expected to reduce alignment.

**Implications.** Taking expectations under this assumption yields

$$\mathbb{E}[\Delta_A] \;>\; 0, \qquad \mathbb{E}[\Delta_B] \;\leq\; 0, \qquad \mathbb{E}[\Delta_C] \;\leq\; \mathbb{E}[\Delta_A]. \tag{22}$$

Hence, Strategy A provides a reliably positive gain by amplifying identified functional heads. Strategies B and C both introduce subtractive terms $(\beta - 1)U(\mathcal{H}_{\text{attenuate}})$, which are non-positive in expectation, making them less stable. This formalizes why, under the *minimal-editing* principle, selective enhancement is the safer choice.

**Takeaways.** Strategy A yields a guaranteed positive gain by enhancing only functional heads. Strategies B and C involve attenuation terms $(\beta - 1)U(\mathcal{H}_{\text{attenuate}})$, which are non-positive in expectation and risk weakening neutral or beneficial heads. Therefore, under the *minimal editing* principle, Strategy A is the most stable and reliable choice.

## C  TIME COMPLEXITY ANALYSIS

In this section, we present a detailed analysis of the algorithmic time complexity of the standard attention mechanism and our proposed method in a single multi-head attention block.

Let the input sequence length be $N$, the model hidden dimension be $d_{model}$, and the number of attention heads be $H$. For each attention head, the query, key, and value dimensions are denoted by $d_k$ and $d_v$, with $d_k = d_v = d_{model}/H$. For notational simplicity, we refer to the per-head dimension as $d_h$. In the following analysis, we focus on the asymptotic complexity with respect to $N$, treating $d_{model}$, $H$, $d_k$, and $d_v$ as constants.

STANDARD ATTENTION

**Input Projections.** Given the input tensor $X \in \mathbb{R}^{N \times d_{model}}$, three linear projections are applied to generate the query, key, and value tensors: $Q = XW_Q$, $K = XW_K$, and $V = XW_V$, where $W_Q, W_K, W_V \in \mathbb{R}^{d_{model} \times d_{model}}$. Each projection is a matrix multiplication with complexity $O(N \cdot d_{model}^2)$, resulting in:

$$T_1 = O(3 \cdot N \cdot d_{model}^2) = O(N \cdot d_{model}^2) \tag{23}$$

**Attention Score Computation.** For each head $h \in \{1, \dots, H\}$, unnormalized attention scores are computed as $Q_h K_h^T$, where $Q_h \in \mathbb{R}^{N \times d_k}$ and $K_h^T \in \mathbb{R}^{d_k \times N}$. The cost per head is $O(N^2 \cdot d_k)$, and across all heads:

$$T_2 = O(H \cdot N^2 \cdot d_k) \tag{24}$$

**Softmax Normalization.** A row-wise softmax is applied to the $N \times N$ score matrix for each head. Applying softmax to a vector of length $N$ costs $O(N)$, and thus to $N$ rows is $O(N^2)$ per head. Across all heads:

$$T_3 = O(H \cdot N^2) \tag{25}$$

**Value Aggregation.** The normalized attention matrix $A_h$ is multiplied by $V_h \in \mathbb{R}^{N \times d_v}$ to produce the head output $O_h \in \mathbb{R}^{N \times d_v}$. The cost per head is $O(N^2 \cdot d_v)$, yielding:

$$T_4 = O(H \cdot N^2 \cdot d_v) \tag{26}$$

**Output Projection.** The $H$ head outputs are concatenated into a $N \times d_{model}$ matrix and projected via $W_O \in \mathbb{R}^{d_{model} \times d_{model}}$ with cost:

$$T_5 = O(N \cdot d_{model}^2) \tag{27}$$

Summing all terms gives the total time complexity:

$$T_{\text{Standard}} = T_1 + T_2 + T_3 + T_4 + T_5 = O(Nd_{model}^2 + HN^2 d_k + HN^2 + HN^2 d_v + Nd_{model}^2)$$

Table 4: Average per-batch inference time (seconds). Parentheses show relative change vs. Vanilla.

| Method | MathVista$_{mini}$ | MathVision$_{mini}$ | HallusionBench | CLEVR | MMStar | SEED-Bench |
|---|---|---|---|---|---|---|
| Vanilla | 98 (+0.0%) | 98 (+0.0%) | 101 (+0.0%) | 74 (+0.0%) | 83 (+0.0%) | 68 (+0.0%) |
| VCD | 159 (+62.2%) | 241 (+145.9%) | 170 (+68.3%) | 116 (+56.8%) | 113 (+36.1%) | 98 (+44.1%) |
| CGD | 482 (+391.8%) | 709 (+623.5%) | 664 (+557.4%) | 314 (+324.3%) | 297 (+257.8%) | 329 (+383.8%) |
| AGLA | 119 (+21.4%) | 164 (+67.3%) | 123 (+21.7%) | 78 (+5.4%) | 87 (+4.8%) | 88 (+29.4%) |
| Ours | **103 (+5.1%)** | **101 (+3.1%)** | **103 (+2.0%)** | **75 (+1.4%)** | **83 (+0.0%)** | **69 (+1.5%)** |

As $N$ grows, the $N^2$ terms dominate. Therefore, the overall asymptotic time complexity is:

$$T_{\text{Standard}} = O(H \cdot N^2 \cdot (d_k + d_v + 1)) = O(N^2) \tag{28}$$

OUR METHOD

**Standard Attention Score and Softmax Computation.** Our method first performs the standard attention computation as in the baseline, obtaining all $H$ normalized attention matrices $A_h \in \mathbb{R}^{N \times N}$:

$$T_1' = O(N d_{model}^2 + H N^2 d_k + H N^2) \tag{29}$$

**Visual Attention Ratio Calculation.** We then compute a visual attention ratio for each head based on Equation 4. This operation traverses all $N \times N$ attention entries, giving:

$$T_2' = O(H \cdot N^2) \tag{30}$$

**Head Categorization and Gain Assignment.** Each head is categorized and assigned a scalar gain $\alpha_h$ based on its $S_v(h)$. This involves only $O(H)$ comparisons and assignments, which are negligible compared to the dominant terms.

**Modulated Value Aggregation.** The values are aggregated as $O_h = A_h V_h$, followed by element-wise scaling with $\alpha_h$. The aggregation part has cost:

$$T_{4a}' = O(H \cdot N^2 \cdot d_v) \tag{31}$$

and the scaling part adds $O(H \cdot N \cdot d_v) = O(N \cdot d_{model})$, which is dominated by the aggregation cost. Thus:

$$T_4' = O(H \cdot N^2 \cdot d_v) \tag{32}$$

**Final Output Projection.** The modulated outputs are concatenated and projected as in the baseline:

$$T_5' = O(N \cdot d_{model}^2) \tag{33}$$

Combining all steps, the total complexity of our method is:

$$T_{\text{Ours}} = T_1' + T_2' + T_4' + T_5' = O(N d_{model}^2 + H N^2 d_k + 2 H N^2 + H N^2 d_v + N d_{model}^2)$$

Focusing on the dominant terms with respect to $N$, we obtain:

$$T_{\text{Ours}} = O(H \cdot N^2 \cdot (d_k + d_v + 2)) = O(N^2) \tag{34}$$

EXPERIMENTS

As reported in Table 4, our method attains per-batch inference times that are essentially indistinguishable from the Vanilla baseline across all six benchmarks, with only marginal fluctuations of a few percentage points. This empirical parity aligns with the theoretical expectation that our additional head-level operations introduce only constant-factor overhead beyond standard multi-head attention. Consequently, the overall time complexity remains unchanged, and in practice, the runtime of our approach matches that of the original model.

## C.1 COMPARISON AND CONCLUSION

Although our method introduces an additional $O(H \cdot N^2)$ step for head-level ratio computation, the total complexity remains dominated by the $O(N^2)$ cost of the standard attention mechanism. As a result, both exhibit the same asymptotic time complexity of $O(N^2)$ with respect to the sequence length $N$. The additional operations only contribute constant-factor overhead without altering the overall computational order.

Consequently, in practice, the runtime of our approach is **essentially indistinguishable** from that of standard attention.

## D IMPLEMENTATION DETAILS

**Ocean-R1 (Ming et al., 2025)** `Ocean-R1` is a large vision-language reasoning model fine-tuned from `Qwen2.5-VL-Instruct` (Bai et al., 2025), designed to enhance cross-modal reasoning and visual understanding capabilities through a two-stage rule-based Reinforcement Learning framework. The first stage focuses on strengthening the model's reasoning ability, while the second stage improves its visual perception. Experimental results demonstrate that `Ocean-R1` achieves substantial performance gains, particularly on visual mathematical reasoning benchmarks such as MathVision (+2.7/+2.7) and MathVista (+4.9/+4.4), showing strong multimodal reasoning and generalization capabilities.

**R1-Onevision (Yang et al., 2025a)** `R1-Onevision` is a state-of-the-art multimodal reasoning model designed to bridge the gap between visual perception and deep reasoning. It is fine-tuned from `Qwen2.5-VL` (Bai et al., 2025), with a focus on cross-modal reasoning that enables precise understanding and processing of both visual and textual information. Unlike previous models that primarily rely on fixed structures for reasoning, `R1-Onevision` employs a two-stage post-training strategy: Supervised Fine-Tuning (SFT) and Reinforcement Learning (RL), enhancing its ability to generalize across diverse tasks. The model leverages a cross-modal reasoning pipeline that transforms images into formal text-based representations, which are then processed to generate structured reasoning paths. It also incorporates a "role-playing" strategy to iteratively refine visual comprehension, ensuring robust multimodal coherence. Experimental evaluations on benchmarks like MathVista and MathVerse demonstrate that `R1-Onevision` outperforms several state-of-the-art models, including GPT-4o and Qwen2.5-VL, showcasing its superior reasoning and generalization capabilities.

**Kimi-VL (Team et al., 2025)** `Kimi-VL` is an efficient open-source vision-language model built upon a Mixture-of-Experts (MoE) language decoder with only 2.8B activated (16B total) parameters, paired with a 400M native-resolution vision encoder (MoonViT). It is designed to provide advanced multimodal reasoning, long-context understanding, and strong agent capabilities, while maintaining high parameter efficiency. Unlike most dense-architecture VLMs, `Kimi-VL` achieves competitive or superior performance to larger models on diverse tasks, including college-level problem solving, OCR, multi-image reasoning, video understanding, and long-document comprehension. Through long Chain-of-Thought supervised fine-tuning and reinforcement learning, its enhanced variant `Kimi-VL-Thinking` demonstrates strong long-horizon multimodal reasoning ability, achieving remarkable results on benchmarks such as MathVision and MathVista. This demonstrates `Kimi-VL`'s effectiveness in combining parameter efficiency with powerful multimodal reasoning capabilities.

Our method is primarily implemented in the `eager_attention_forward` function within the modeling file (`modeling_qwen2_5_vl.py` and `modeling_kimi_vl.py`). Also, we incorporate a caching mechanism to store essential information (e.g., the range of visual tokens and hyperparameters).

Table 5 summarizes the hyperparameters used in our experiments.

## E DATASETS

**MathVista (Lu et al., 2023)** Aggregates 6,141 problems by consolidating 28 existing multimodal datasets with three newly curated sources—IQTest (puzzle figures), FunctionQA (function plots), and PaperQA (figures from academic papers)—spanning charts, diagrams, textbook geometry, and

Table 5: Selected Hyperparameters

| Model | $\ell_{\text{reas}}$ | $\tau_{\text{reas}}$ | $g_{\text{reas}}$ | $\ell_{\text{perc}}$ | $\tau_{\text{perc}}$ | $g_{\text{perc}}$ |
|---|---|---|---|---|---|---|
| Kimi−VL−A3B−Thinking | 5 | 0.01 | 1.40 | 10 | 0.27 | 1.20 |
| Ocean−R1−7B−Instruct | 3 | 0.01 | 1.30 | 7 | 0.22 | 1.16 |
| R1−Onevision | 3 | 0.01 | 1.30 | 7 | 0.30 | 1.20 |

everyday VQA scenes. It targets cross-domain mathematical reasoning grounded in images, with compositional perception and minimal leakage, and remains challenging for current models.

**MathVision** (Wang et al., 2024b) Curated from real math competitions to provide 3,040 image-based problems across 16 mathematical disciplines (e.g., analytic geometry, topology, graph theory) and five difficulty levels. Designed to stress rigorous multimodal mathematical reasoning beyond earlier benchmarks, it exhibits a large human–model gap and offers *test* and *test-mini* splits for rapid benchmarking.

**MMStar** (Chen et al., 2024) Built by screening items from existing benchmarks and then manually vetting them to ensure vision indispensability and remove data leakage, yielding 1,500 human-selected samples covering six core capabilities along 18 axes. Its goal is a purified, balanced test that measures true multimodal gains—i.e., questions where visual evidence is necessary and difficulty stems from advanced cross-modal reasoning.

**CLEVR** (Johnson et al., 2017) A synthetic diagnostic VQA dataset rendered in Blender with simple 3D objects varying in color, shape, size, and material; questions are programmatically generated and paired with functional programs and scene graphs. The design emphasizes low dataset bias and compositional reasoning difficulty (counting, comparisons, spatial relations) over 100k images and 1M questions.

**HallusionBench** (Guan et al., 2024) Comprises 346 figures from diverse sources and formats (e.g., charts, tables, maps, famous optical illusions, memes) paired with 1,129 expert-authored questions structured into control pairs and "visual-dependent" vs. "visual-supplement" settings. It is purpose-built to disentangle language hallucination from visual illusion and to test consistency under easy/hard conditions, remaining challenging for state-of-the-art LVLMs.

**SEED-Bench** (Li et al., 2024) Uses human-verified multiple-choice questions targeted to capability dimensions; in SEED-Bench-1 (19k), images come from Conceptual Captions and videos from Something-Something V2, Epic-Kitchens, and Breakfast, covering both spatial and temporal understanding. The benchmark's design goal is dimension-specific difficulty and objective automatic scoring across 12 image/video comprehension dimensions.

# F  BASELINES

We adopt a deliberately diverse set of baselines. Methods for mitigating inference-time hallucinations via generation intervention can be grouped into three families: **Contrastive Decoding**, **Guided Decoding**, and **Visual Amplification**. Accordingly, we select VCD, CGD, and AGLA as representative instances. These baselines instantiate three complementary mechanisms—probability calibration, external guidance, and feature enhancement—covering different failure modes while reducing evaluation bias due to methodological homogeneity; together they constitute a representative comparison suite.

**Visual Contrastive Decoding (VCD)** (Leng et al., 2024). VCD is a training-free decoding strategy that explicitly builds a counterfactual view of the image by introducing controlled visual uncertainty and then contrasts the model's predictions under the original versus the uncertain view. Tokens that are preferred only when the image is ambiguous are down-weighted, which directly combats two root causes of object hallucination: over-reliance on language priors and spurious object co-occurrence. An adaptive plausibility constraint further preserves fluent generation by truncating to high-confidence candidates from the original distribution. The key difference from other baselines is that VCD does not require external models or retraining; it modifies only the sampling distribution and therefore generalizes across LVLM families.

**Assembly of Global and Local Attention (AGLA) (An et al., 2025).** AGLA targets the attention deficiency behind many hallucinations: LVLMs tend to lock onto prompt-irrelevant global patterns while missing fine-grained, prompt-relevant regions. It first performs image–prompt matching (via a Grad-CAM–style analysis) to produce an augmented view that highlights regions tied to the query and suppresses distractors; decoding then fuses evidence from the global view (original image) and the local, discriminative view (augmented image). This design differs from VCD and CGD by enriching the *visual features* themselves rather than only reshaping probabilities, which improves grounding without sacrificing generative context. Empirically, it is especially strong under challenging negative settings, e.g., achieving about 81.36 F1 on the adversarial split of POPE with LLaVA-1.5.

**CLIP-Guided Decoding (CGD) (Deng et al., 2024).** CGD injects an external, vision-language prior—CLIP—directly into decoding, but at the *sentence level*. Instead of judging partial tokens, CGD scores complete sentence candidates by mixing the LVLM's length-normalized likelihood with CLIP image–text similarity and keeps candidates that are both probable and visually grounded. Two design choices explain its gains: (i) sentence-level guidance avoids the myopia of token-level heuristics and directly penalizes late-caption drift (later sentences are empirically more hallucinatory), and (ii) CLIP similarity provides a stronger, more stable signal of image–text alignment than model likelihood alone. As a result, CGD substantially lowers hallucination while preserving caption quality, e.g., CHAIRs on COCO with LLaVA-1.5 drops from 44.7 to 29.7.

## G  METRICS

We adopt the *F1 score* as the primary evaluation metric, with calculation logic adapted to the nature of each dataset. Specifically, two distinct evaluation schemes are applied: (1) a binary judgment scheme for datasets containing binary ground-truth labels (e.g., HallucinationBench), and (2) a multi-class scheme for datasets containing multiple-choice questions (e.g., MMStar and SEED-Bench). In both schemes, the final performance is reported using the *Weighted F1 score*.

**Binary Judgment Scheme**  For binary datasets, we use the standard F1 score, defined as:

$$F1 = \frac{2 \times \text{Precision} \times \text{Recall}}{\text{Precision} + \text{Recall}} = \frac{2TP}{2TP + FP + FN}. \tag{35}$$

**Multi-class Scheme**  For multi-class datasets, each sample has a ground-truth label $y \in \{1, \ldots, K\}$ and a predicted label $\hat{y} \in \{1, \ldots, K\}$. For each class $i \in \{1, \ldots, K\}$, we define:

$$TP_i = |\{\hat{y} = i, \, y = i\}|, \qquad FP_i = |\{\hat{y} = i, \, y \neq i\}|, \tag{36}$$
$$FN_i = |\{\hat{y} \neq i, \, y = i\}|, \qquad TN_i = |\{\hat{y} \neq i, \, y \neq i\}|. \tag{37}$$

Then:

$$\text{Precision}(i) = \frac{TP_i}{TP_i + FP_i}, \qquad \text{Recall}(i) = \frac{TP_i}{TP_i + FN_i}, \tag{38}$$

$$F1(i) = \frac{2 \times \text{Precision}(i) \times \text{Recall}(i)}{\text{Precision}(i) + \text{Recall}(i)}. \tag{39}$$

**Weighted Aggregation**  To account for class imbalance, the weighted F1 score is computed as:

$$\text{Weighted-F1} = \sum_{i=1}^{K} \frac{\text{Support}(i)}{N} \times F1(i), \tag{40}$$

where $\text{Support}(i)$ denotes the number of samples with ground-truth label $i$, and $N$ is the total number of samples.

## H  CASE ANALYSIS

### H.1  CONTRIBUTION MAP

Following Michel et al (Michel et al., 2019)., we quantify the contribution of each attention head to a *specific next-token prediction* by inserting differentiable gates on head outputs and reading out

the loss gradients w.r.t. these gates. Intuitively, if slightly amplifying head $(l, h)$ would decrease the loss for the current target token, that head is *helpful* for this token; if it would increase the loss, it is *harmful*. We report absolute sensitivities as non-negative importance scores and optionally keep the signed scores to analyze supportive vs. adversarial heads.

**Notation at a glance.**

- $B$: batch size; $S$: sequence length seen by the model at the current decoding step (includes prompt and previously generated tokens).
- $L$: number of transformer layers.
- $H$: number of attention heads per layer.
- $d_{\text{model}}$: hidden size; $d_{\text{head}} = d_{\text{model}}/H$: per-head width after splitting.
- $\mathbf{x}$: model inputs (encoded sequence including both image and text); $\mathbf{y}_{<\tau}$: generated prefix; $y_\tau$: random variable at current position; $t^\star$: target token for this step.
- $\mathcal{L} = -\log P(y_\tau = t^\star \mid \mathbf{x}, \mathbf{y}_{<\tau})$: cross-entropy loss defined only for current position $\tau$.

### H.1.1 GATE PARAMETER MECHANISM

We insert lightweight gate parameters onto each attention head to directly modulate their outputs without changing model weights. This setup allows us to treat each gate as a differentiable scalar that controls the influence of its head. With this mechanism in place, we can measure how scaling a head affects the model's loss for the current token.

For layer $l$, let the standard multi-head output (after attention and output projection, before the MLP) be

$$\mathbf{O}^{(l)} \in \mathbb{R}^{B \times S \times d_{\text{model}}}. \tag{41}$$

We reshape to expose heads:

$$\mathbf{O}^{(l)}_{\text{heads}} = \text{reshape}\left(\mathbf{O}^{(l)}, [B, S, H, d_{\text{head}}]\right). \tag{42}$$

We use a trainable-on-the-fly (but *not optimized*) gate vector $\mathbf{g}^{(l)} \in \mathbb{R}^H$ initialized as $\mathbf{1}_H$ and broadcast it multiplicatively:

$$\forall b \in [B],\ s \in [S],\ h \in [H]: \quad \mathbf{O}^{(l)}_{\text{gated}}[b, s, h, :] = g^{(l)}_h \mathbf{O}^{(l)}_{\text{heads}}[b, s, h, :]. \tag{43}$$

Finally we fold heads back to $d_{\text{model}}$ and continue the standard forward. Implementation uses lightweight forward hooks; model weights remain frozen. Gates are created with `requires_grad=True` so that backprop can produce $\frac{\partial \mathcal{L}}{\partial g^{(l)}_h}$.

**Shape sanity check.** $\mathbf{O}^{(l)}_{\text{heads}}[b, s, h, :] \in \mathbb{R}^{d_{\text{head}}}$ is the contribution of head $h$ at token position $s$. Multiplying by $g^{(l)}_h$ scales the entire vector produced by that head uniformly at all positions in the current pass; this matches the "masking/scale" abstraction in Michel et al.

### H.1.2 GRADIENT-BASED IMPORTANCE

By backpropagating the cross-entropy loss through these gates, we obtain per-head gradients that quantify how loss would change if a head were amplified. Negative gradients indicate supportive heads, while positive gradients reveal harmful ones. Taking the absolute values yields stable importance scores for both visualization and analysis.

At decoding step $\tau$ with target $t^\star$,

$$\mathcal{L} = -\log P(y_\tau = t^\star \mid \mathbf{x}, \mathbf{y}_{<\tau}). \tag{44}$$

We define the *signed sensitivity* and the *importance* for head $(l, h)$:

$$S^{(l,h)} = \frac{\partial \mathcal{L}}{\partial g^{(l)}_h}, \qquad I^{(l,h)} = \left| S^{(l,h)} \right|. \tag{45}$$

Interpretation: a negative $S^{(l,h)}$ means "if we scale up this head, loss goes down" (supportive head); a positive $S^{(l,h)}$ suggests the opposite (potentially harmful). Reporting $|S|$ yields non-negative "importance heatmaps", while retaining the sign of $S$ allows for more fine-grained adversarial analysis.

**Connection to first-order Taylor pruning.** Expanding $\mathcal{L}$ at $\mathbf{g} = \mathbf{1}$, a small perturbation $\Delta g_h^{(l)}$ changes the loss by $\approx S^{(l,h)} \Delta g_h^{(l)}$; therefore $|S^{(l,h)}|$ can directly serve as a "influence measure for current sample/step", which is the per-sample instantiation of the expected sensitivity $I_h = \mathbb{E}\,|\partial \mathcal{L}/\partial \xi_h|$ proposed by Michel et al. under our gate parameterization.

### H.1.3 FROM PER-STEP SCORES TO A RANKING

Finally, we aggregate these per-head importance scores and normalize them either layer-wise or globally. Sorting the normalized scores produces a ranking that highlights which heads are most influential for the given token (or across tokens when averaged). This ranking provides a principled basis for head attribution and comparison.

Collect raw scores

$$\mathcal{I} = \{I^{(l,h)} \mid l = 1, \ldots, L; h = 1, \ldots, H\}. \tag{46}$$

We consider two normalizations:

**Layer-wise** (robust to layer-scale differences):

$$\tilde{I}_{\text{layer}}^{(l,h)} = \frac{I^{(l,h)}}{\sum_{h'=1}^{H} I^{(l,h')}}. \tag{47}$$

**Global** (for global ranking across the whole stack):

$$\tilde{I}_{\text{global}}^{(l,h)} = \frac{I^{(l,h)}}{\sum_{l'=1}^{L} \sum_{h'=1}^{H} I^{(l',h')}}. \tag{48}$$

We then sort

$$\pi = \text{argsort}\big(\tilde{I}, \text{ descending}\big), \tag{49}$$

obtaining a head ranking tailored to the current token $t^\star$ (or to an aggregate of tokens; see below).

### H.2 CASE I: REASONING DRIFT

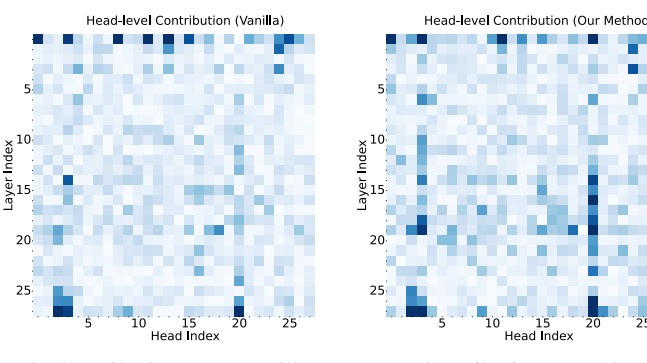

We welcome change and openness; for we believe that freedom and security go together, that the advance of human liberty can only strengthen the cause of world peace. There is one sign the Soviets can make that would be unmistakable, that would advance dramatically the cause of freedom and peace. General Secretary Brezhnev, if you seek peace, if you seek prosperity for the Soviet Union and Eastern Europe, if you seek liberalization: Come here to this gate! Mr. Brezhnev, open this gate! Mr. Brezhnev, tear down this wall!

(a) Question        (b) Contribution Map (Vanilla)        (c) Contribution Map (Ours)

> **Question - Image (a)**
>
> Is the text in this figure trying to talk to Gorbachev?

**Vanilla - Image (b)**

<think>
Okay, so I need to figure out whether the text in the image is trying to talk to Gorbachev. Let's start out by understanding the context. The image shows a text message or a letter. The text is talking about change and openness and mentions General Secretary Brezhnev. So, Brezhnev was the leader of the Soviet Union during the Cold War. Gorbachev came later, so he might have been the General Secretary later in the 80s.

Looking at the text, it says things like "Come here to this gate!" and "Tear down this wall!". These phrases sound like they're addressing someone specific, probably someone in power authority, which would be the General Secretary. The image also has uses the term "General Secretary Brezhenve', which is a typo for Brezhnev. This reinforces the idea that the speaker is talking to someone in authority.

Also, the message is urgent to to the cause of freedom and peace and economic prosperity in Eastern Europe. This kind of plea would be directed at someone who is in power, like a leader. Gorbachev was famous for bringing Soviet policies and for making reforms like glasuez, which aimed at openness and change in the Soviet Union. So, it makes sense that this message is addressed to him.

The image shows a letter by someone urging change and openness, asking Brezhnev to do something, and even mentioning tearing down a wall. All these points to in message intended at Gorbachev. The reference to Brezhnev being the leader, even if the name spelling is incorrect, points to the main target of the message. So, yes, the text is trying to talk to Gorbachev.
</think>

The text in the image is clearly directed at Gorbachev. The mention of General Secretary Brezhnev, the reference to tearing down a wall, and the plea for change and openness all point to in address to Gorbachev, who a leader known for the reforms aimed at opening in in Eastern Europe. Therefore, the message is addressed to Gorbachev.

**Answer:** **Yes**, the text in the image is trying to talk to Gorbachev.

---

**Ours - Image (c)**

<think>
Okay, so I'm trying to determine if the text is trying to talk to Gorbachev. Let's start with the image directly.

The image shows a request to open the gates. However, the text in the image is addressed to General Secretary Brezhnev. The image itself depicts a scene where Gorbachev might be the leader of the Soviet Union. The image also a reference to " Gorbachev, which gives a hint that the leader might Gorbachev's the leader of the Soviet Union.

The image shows a request to open the gates. The text in the image is addressed to General Secretary Brezhnev. The image itself shows a scene where Gorbachev might be the leader of the Soviet Union. The image also refers to Gorbachev, but the text is addressed to Brezhnev.

Therefore, the text is not directly addressing Gorbachev; it is instead addressed to Brezhnev, the leader of the Soviet Union at the time.
</think>

The image shows a request to open the gates. Therefore, the text is not trying to talk to Gorbachev; it is trying to talk to Brezhnev, the leader of the Soviet Union at the time.

**Answer:** No, the text in the image is not trying to talk to Gorbachev; it is referring to Brezhnev, the leader of the Soviet Union at the time.

**Analysis.** From the contribution maps, our method markedly amplifies the mid-to-deep *reasoning heads* in Layer 10–23, thereby substantially improving the model's reasoning capacity. In contrast, the Vanilla model shows no obvious perceptual failure but suffers during reasoning: because the question mentions "Gorbachev", Vanilla model erroneously binds this cue to the entire message, accumulates evidence under the hypothesis "target = Gorbachev," and drifts away from the correct logical chain, ultimately misclassifying an appeal addressed to Brezhnev as speaking to Gorbachev. By strengthening these mid-to-deep reasoning heads, our method maintains stable entity grounding and a consistent causal narrative, **preventing reasoning drift** and producing the correct negative answer.

### H.3 CASE II: PERCEPTUAL BIAS

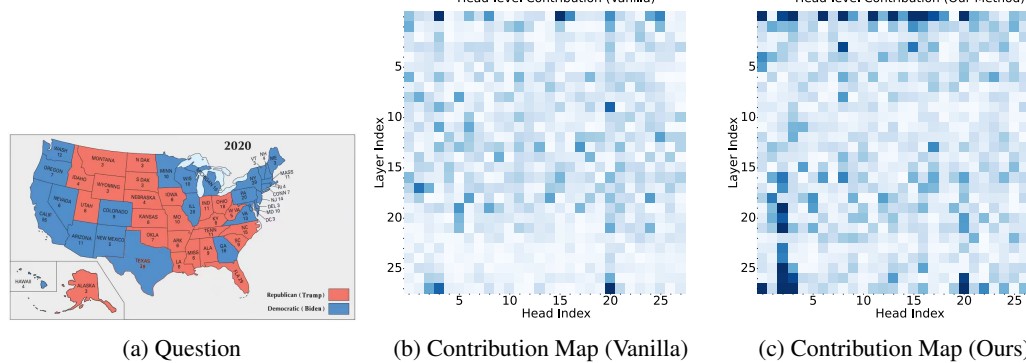

(a) Question        (b) Contribution Map (Vanilla)        (c) Contribution Map (Ours)

---

**Question - Image (a)**

Based on the map, did the Republic Party win Texas in the 2020 elections?

---

**Vanilla - Image (b)**

<think>
Okay, so I'm trying to determine whether the Republican Party won Texas in the 2020 elections based on the provided map. Let me break this down step.

First, the image shows a map of the United States indicating the election results by state for the 2020 U.S. presidential election. Each state is colored either **red** or blue, representing Republican and Democratic wins, respectively. The image shows Texas colored in **red**.

To confirm, the image legend has the **red** color for Republican (Trump) and blue for Democratic (Biden). Therefore, since Texas is colored **red**, it indicates that the Republican Party won Texas in the 2020 election.

There are no other states mentioned that would affect this conclusion. The other states might be colored differently, but Texas being red is the key point of this question.

So, putting it all, the image directly shows Texas as **red**, which means the Republican Party won Texas in the 2020 election.
</think>

The image shows Texas colored **red**, which indicates that the Republican Party won Texas in the 2020 election.

**Answer:** **Yes**, the Republican Party won Texas in the 2020 election.

---

Ours - Image (c)

<think>
Okay, so I'm trying to determine whether the Republican Party won Texas in the 2020 elections based on the provided map. Let me go through this step by step.

First, I need to understand what the map is showing. It is an electoral map from the 2020 U.S. Presidential Election, where each state is colored either **red** or blue. **Red** represents Republican candidates, and blue represents Democratic candidates. Each state also have a number 2020, so this is the 2020 election results.

Next, I notice that Texas is colored blue. This indicates that Texas was won by the Democrats. Therefore, the Republican Party did not win Texas in the 2020 elections.
</think>

The answer is that the Republican Party did not win Texas in the 2020 elections. Texas was won by the Democrats.

---

**Analysis.** In this case, the contribution maps reveal that our method effectively mitigates the dispersed and unfocused attention observed in the Vanilla model by strengthening both shallow perception heads and deep reasoning heads. The primary issue with the Vanilla model arises in the second paragraph of its reasoning, where it misperceives the color of Texas as red. This initial perceptual bias propagates through its logical reasoning process, ultimately leading to an incorrect conclusion that the Republican Party won Texas. In contrast, our method, supported by **stronger perceptual attention** in shallow layers and more robust reasoning capacity in deeper layers, accurately perceives Texas as blue and correctly infers that the state was won by the Democrats. This demonstrates how our approach enhances the interplay between perception and reasoning, enabling the model to avoid compounding errors and to arrive at the correct answer.

## H.4 CASE III: COMPREHENSIVE HALLUCINATION

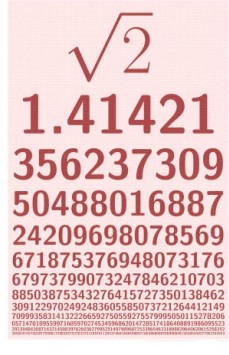

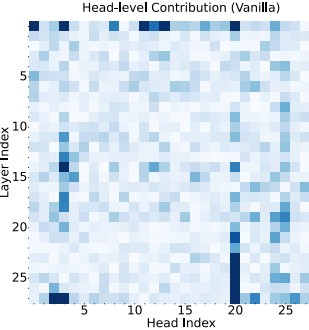

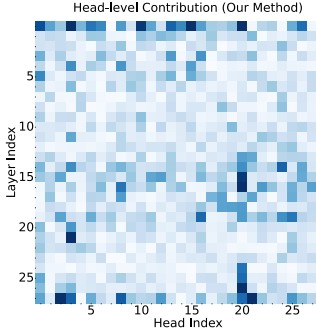

(a) Question      (b) Contribution Map (Vanilla)      (c) Contribution Map (Ours)

---

Question - Image (a)

According to the image, does the value of the square root of 2 range from 1.40 to 1.41?

---

> **Vanilla - Image (b)**
>
> <think>
>
> Okay, so I need to determine whether the square root of 2 ranges from 1.40 to 1.41 based on the image provided. The image shows several square roots, and I need to identify the square root of 2.
> 1. As the image shows several square roots, and the first one 2 is approximately 1.44.
> 2. The image also shows the square root of 2 as approximately 1.44.
> 3. Therefore, the image shows the square root of 2 as approximately 1.44.
>
> </think>
>
> Answer: The square root of 2 is not between 1.0 and 1.1; it is approximately 1.4.

> **Ours - Image (c)**
>
> <think>
> Okay, so I need to figure out whether the value of the square root of 2, denoted as sqrt(2), falls within the range of 1.40 to 1.41. Let's start directly based on the image and the information provided.
>
> The image shows the square root of 2 as approximately **1.41421**. The image also shows various mathematical constants and numbers sequences, including the first 150 decimal places, and it provides many of the first decimal places as 1.4 ...
> </think>
> Answer: No. It's about 1.414, which is slightly above 1.41.

**Analysis.** In this case, we can clearly observe the differences in perception and reasoning abilities between the two methods. The Vanilla model, when tasked with identifying the value of the square root of 2, incorrectly perceived it as 1.44 rather than the correct value of 1.414. This perceptual error indicates insufficient contribution from the shallow perception heads, leading to poor sensitivity to numerical details. Furthermore, the subsequent reasoning process in the Vanilla model was unclear, with its step-by-step logic lacking coherence, and the final answer deviated from the original question. This combination of perceptual inaccuracy and weak reasoning aligns with the contribution map, which shows limited activation in both shallow perception heads and mid-to-deep reasoning heads. In contrast, our method demonstrates significant enhancements in both shallow perception heads and mid-to-deep reasoning heads, resulting in a more balanced improvement across perception and reasoning capabilities. Specifically, in this example, our method not only correctly identified the square root of 2 as 1.41421 but also precisely reasoned that the value is slightly above 1.41, thereby providing an accurate and contextually relevant answer. This highlights the superiority of our approach in achieving **improvements in both perceptual accuracy and logical reasoning**.

## H.5 CASE IV: COMPREHENSIVE HALLUCINATION

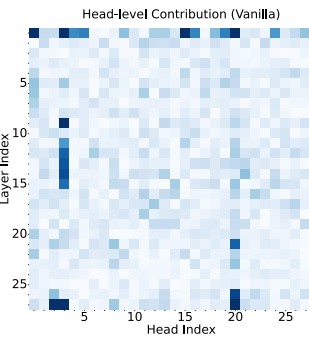
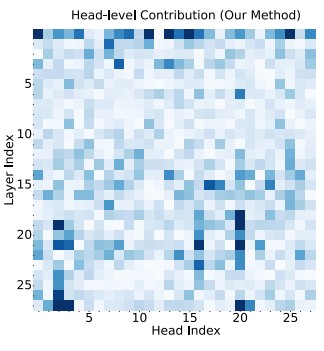

(a) Question      (b) Contribution Map (Vanilla)      (c) Contribution Map (Ours)

---

**Question - Image (a)**

There is a big thing that is the same color as the tiny ball; what shape is it?

---

**Vanilla - Image (b)**

<think>
Okay, so I'm trying to solve this problem where I need to figure out the shape of a big object that's the same color as a tiny ball. Let's start with understanding the question carefully. The word "same color" probably refers to the same color group, meaning, or category, but I think it might be a bit too complicated.

The image shows a 3D geometric composition with a large red sphere, which is the " same color" of the small red sphere. Hmm, so the question is asking for the shape of a large object that's's the same color as the tiny ball. So the tiny ball is a sphere, and the large one the same color as the tiny ball would also be a sphere as well, but much bigger in size.

The image shows a large red sphere within other shapes, so the large red sphere stands out the same color of another 3D object. The only other 3D object that could be described as the same color would also be another sphere. The image shows a large red sphere, where another 3D object can be a sphere. Therefore, the large red sphere stands out the same color of another sphere, which would also be a sphere as well. So, the large red sphere is a sphere, and another 3D object sharing the same color can be a sphere as well.

So the answer should be a sphere.
</think>

The image shows a large red sphere, which is the same color of another sphere. Therefore, the large red sphere is a sphere, and the one same color would be a sphere as well. The image shows a large red sphere, with another shape being a large red sphere. So, the large red sphere is a sphere, and another 3D object can be a sphere as well.

**Answer:** Sphere

---

**Ours - Image (c)**

<think>
Okay, so I'm trying to figure out the answer to this question: There is a big thing that is the same color as the tiny ball; what shape is it?

The image shows a big red cube next to a tiny red sphere. The tiny ball is a sphere, and the big thing mentioned is a red cube. Since the image shows a big red cube next to a tiny red sphere, the big thing must be a red cube.

Therefore, the answer is a red cube.
</think>

The image shows a big red cube next to a tiny red sphere. The tiny ball is a sphere, and the big thing mentioned is a red cube.

Answer: The shape is a red cube.

**Analysis.** The Vanilla model's reasoning exhibits systemic failure: it is convoluted, opaque, and internally inconsistent. Contribution maps reveal diffuse attention without stable anchoring, producing weak and unreliable signals across both perceptual and relational heads. More critically, attribute cross-talk collapses the distinction between color and shape, preventing the model from executing the necessary reasoning pipeline of anchoring $\rightarrow$ same-color filtering $\rightarrow$ applying the "big" constraint. As a result, it defaults to a spurious shortcut and outputs the incorrect label "sphere." In contrast, our method imposes structured coordination: shallow heads focus sharply on color and size cues, while deep heads strengthen cross-object reasoning that jointly enforces the predicates "same color as the tiny ball" and "big," leading to the correct identification of the large red cube. These results highlight that without principled alignment between perception and reasoning, models are prone to brittle and misleading inference.

