# OpenReview forum: "Mitigating Hallucination in Multimodal Reasoning via Functional Attention Control"
_ICLR.cc/2026/Conference — ICLR 2026 Conference Withdrawn Submission_

### Official Review · Reviewer_JKvq · 2025-10-21

**Soundness:** 2
**Presentation:** 3
**Contribution:** 2
**Rating:** 2
**Confidence:** 5

**Summary:**

This paper investigates the functional roles of attention heads in multimodal large reasoning models, particularly in perception and reasoning tasks. The authors analyze attention patterns to categorize heads as perception-oriented or reasoning-oriented, based on their focus on visual or textual modalities. Building on the observation that early heads tend to focus on images while later ones attend more to text, they propose a steering strategy that rebalances head attention through a plug-and-play rescaling method, without retraining the model.

**Strengths:**

- **Comprehensive empirical study**: The paper conducts an extensive evaluation covering over 150 boundary configurations and 24 scaling strategies, across 6 benchmarks, 4 model baselines, and multiple multimodal reasoning tasks, offering a robust and reproducible analysis.

- **Practical and efficient method**: The proposed rescaling mechanism is lightweight, easily applicable, and does not require retraining, making it a low-cost yet effective plug-and-play improvement.

- **Grounded in prior findings**: The analysis builds upon established observations that visual heads are typically concentrated in earlier layers, while reasoning heads emerge in later ones, and explores how this structure can be exploited for targeted steering.

**Weaknesses:**

- **Questionable modality-based interpretation in deeper layers**: While early layers remain largely modality-specific, intermediate and late layers increasingly encode multimodal, contextually grounded representations. Therefore, the paper’s use of a modality attention ratio as a proxy for quantifying visual versus textual information becomes problematic beyond early layers.

- **Oversimplified perception–reasoning dichotomy**: Following the previous point, the assumption of a strict split between perception (visual) and reasoning (textual) is conceptually fragile. Reasoning processes in later layers are often driven by early visual activations, making it inaccurate to directly associate perception with image tokens and reasoning with text tokens. The proposed binary segmentation of layers overlooks these interdependencies. Observations 5 and 6 in the paper also seem inconsistent with a universal two-block division, suggesting that the initial assumption may need refinement. A more rigorous investigation, such as testing perception boundaries within reasoning layers and vice versa, could reveal whether this dichotomy truly holds.

- **Imbalanced token contributions**: Since image tokens typically outnumber text tokens, summing attention values across modalities without re-weighting introduces a bias. The decision to use raw sums rather than normalized values by token count should be better justified.

- **Choice of scaling layers insufficiently analyzed**: The authors mention that re-scaling at one layer propagates through subsequent layers (lines 282–283). If so, the choice of which layer(s) to scale becomes crucial, yet this factor is not systematically explored.

- **Neglect of dataset-related confounds (e.g., data leakage)**: A major limitation of MLRMs is data leakage, which can bias models toward reasoning shortcuts rather than perception. Investigating how reasoning heads behave under such conditions could strengthen the perception–reasoning framework and validate whether their steering approach mitigates these known failure modes.

**Questions:**

1. How are unlabeled or ambiguous heads interpreted in your framework? Beyond perception and reasoning, could there be other functional categories?

2. Since rescaled activations do not sum to 1 after applying gains, could this unnormalized behavior introduce instability? Why not re-normalize attention distributions post-scaling?

3. How does the perception–reasoning boundary change with model depth? For deeper or shallower models, does one type of block dominate, and does this affect sensitivity to visual or linguistic features?

---

### Official Review · Reviewer_8qxA · 2025-10-30

**Soundness:** 3
**Presentation:** 3
**Contribution:** 2
**Rating:** 4
**Confidence:** 3

**Summary:**

This paper proposes a plug-and-play, training-free, architecture-agnostic method to mitigate hallucinations in multimodal large reasoning models by selectively enhancing “functional” attention heads associated with perception and reasoning. The authors argue that hallucination arises from both perceptual bias in shallow layers and reasoning drift in deeper layers, and thus introduce selective head-control policies that amplify helpful heads at inference time without modifying model parameters. The method shows improvements across several benchmarks while maintaining efficiency and interpretability.

**Strengths:**

### Strengths
1. The paper identifies two distinct forms of hallucination (perceptual bias and reasoning drift) and attributes them to layer-wise attention dynamics, aligning with recent interpretability literature.

2. The method operates purely at inference and does not require retraining or structural modification, improving deployment practicality.

3. The paper offers a formal treatment of head-scaling strategies and analyzes why selective enhancement is preferable to attenuation-based interventions.

4. Multiple datasets and models are tested, and the head-importance visualizations offer explanatory clarity that aligns method with interpretability goals.

**Weaknesses:**

1.    While the framing and theoretical perspective are valuable, the core idea—adjusting attention gating based on functional signals—is conceptually close to emerging inference-time hallucination mitigation and attention routing techniques. The paper would benefit from a clearer differentiation from works such as inference-time re-attention / attention steering methods or visual retracing-based approaches.

2.    The method’s performance under long-context CoT, dense visual reasoning, and highly noisy images is not thoroughly evaluated. Real-world complexity may affect stability of head-based selection.

3.    It is unclear whether identified “good heads” generalize across architectures and tasks or require task-specific calibration. More systematic robustness or transfer-analysis would strengthen claims.

4.    Although improvement is shown, examples where head-amplification misleads reasoning (e.g., reinforcing biased heads) are not provided.

**Questions:**

1. How consistent are the selected heads across different datasets and vision-domains?
2. Does the method remain effective on long multi-turn reasoning or dense scene understanding tasks?
3. Are there documented negative cases where selective enhancement amplifies spurious correlations?
4. How sensitive is performance to the proportion of heads enhanced?
5. Can the approach support streaming/faster-decoding setups without full forward access to internals?

---

### Official Review · Reviewer_iyPV · 2025-10-31

**Soundness:** 2
**Presentation:** 3
**Contribution:** 3
**Rating:** 4
**Confidence:** 4

**Summary:**

The paper argues that the hallucination for multimodal large reasoning models is largely due to insufficient leverage of perception- and reasoning-oriented heads. Therefore, it proposes a two-step method, functional head identification and class-conditioned rescaling, to mitigate the hallucination.

**Strengths:**

1. The paper is well written and easy to understand.

2. The method of identifying and re-weighting the perception- and reasoning-oriented heads is novel and useful.

3. The proposed method outperforms the baseline and other methods consistently across 6 benchmarks.

4. The work conducts sufficient ablation studies to verify the effectiveness and robustness of the method.

**Weaknesses:**

1. This paper aims to mitigate hallucination in multimodal reasoning models. However, the method is evaluated on only one hallucination-related benchmark (HallucinationBench), which is not enough. You should add evaluations on other hallucination benchmarks like POPE, CHAIR, and MMHal-Bench.

2. The hyperparameters of this work require a lot of hyperparameter search to get a proper configuration. The hyperparameters are dependent on the model architectures, the models themselves, and even tasks. For example, in the provided codebase, different configurations are used for Kimi-VL and Qwen2.5-VL-based models. This makes it difficult for the method to transfer to new scenarios.

3. There is no strong evidence to support the decomposition of hallucinations in MLRMs into perceptual bias and reasoning drift.

**Questions:**

1. Could you provide more evaluation results on hallucination-related benchmarks, such as POPE, CHAIR, and MMHal-Bench?

2. Why are the improvements in mathematical reasoning and multimodal integration in Table 1 related to hallucination mitigation?

3. Could you provide more information about the cost when applying the method to new scenarios (e.g., new model architectures)?

4. Why do you identify the perceptual bias and reasoning drift as two primary failure causes in MLRMs' hallucination?

5. Your methods amplify the visual perception ability and symbolic thinking ability of the multimodal models. Can this method be transferred to non-hallucination tasks and non-thinking multimodal models?

---

### Note · Authors · 2025-11-12

**Comment:**

We would like to withdraw this submission due to internal considerations.
We sincerely appreciate the time and effort of the reviewers and the program committee.

**Withdrawal Confirmation:**

I have read and agree with the venue's withdrawal policy on behalf of myself and my co-authors.